# Foraminiferal and Palynological Records of an Abrupt Environmental Change at the Badenian/Sarmatian Boundary (Middle Miocene): A Case Study in Northeastern Central Paratethys

Danuta Peryt [1], Przemysław Gedl [2], Elżbieta Worobiec [3], Grzegorz Worobiec [3] and Tadeusz Marek Peryt [4,*]

1 Institute of Paleobiology, Polish Academy of Sciences, Twarda 51/55, 00-818 Warszawa, Poland; d.peryt@twarda.pan.pl
2 Institute of Geological Sciences, Polish Academy of Sciences, Research Centre in Kraków, Senacka 1, 31-002 Kraków, Poland; ndgedl@cyf-kr.edu.pl
3 W. Szafer Institute of Botany, Polish Academy of Sciences, Lubicz 46, 31-512 Kraków, Poland; e.worobiec@botany.pl (E.W.); g.worobiec@botany.pl (G.W.)
4 Polish Geological Institute—National Research Institute, Rakowiecka 4, 00-975 Warszawa, Poland
* Correspondence: tadeusz.peryt@pgi.gov.pl

**Abstract:** The Badenian/Sarmatian boundary in the Central Paratethyan basins is characterised by a change from open marine conditions during the late Badenian to the assumed brackish conditions during the early Sarmatian. The foraminiferal and palynological results of the Badenian/Sarmatian boundary interval in the Babczyn 2 borehole (in SE Poland) showed that the studied interval accumulated under variable, unstable sedimentary conditions. The Badenian/Sarmatian boundary, as correlated with a sudden extinction of stenohaline foraminifera, is interpreted as being due to the shallowing of the basin. The lack of foraminifera and marine palynomorphs just above the Badenian/Sarmatian boundary can reflect short-term anoxia. The composition of the euryhaline assemblages, characteristic for the lower Sarmatian part of the studied succession, indicates from marine to hypersaline conditions.

**Keywords:** foraminifera; palynofacies; dinocysts; pollen; spores; Miocene; Central Paratethys; Carpathian Foredeep; Poland

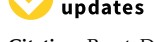



## 1. Introduction

The largest faunal turnover event in the Paratethys—a large epicontinental sea extending from Central Europe to inner Asia since the Oligocene [1]—occurred in the Middle Miocene at the Badenian/Sarmatian boundary, when 94% of Badenian species became extinct (Badenian–Sarmatian Extinction Event—BSEE; [2]). It was proposed that the cause of this event was a drastic basin-wide change in water chemistry from open marine conditions during the late Badenian to brackish conditions in the Central Paratethyan basins during the early Sarmatian. The details of this transformation are still under discussion, though [3,4] concluded that there is no evidence of shallowing associated with the BSEE. Ref. [5] suggested that because of a complex tectonic evolution, the closure of the Paratethyan basins at the end of Badenian was most probably diachronous, and, thus, slightly different ages for the Badenian/Sarmatian boundary across the Paratethys should be expected.

This study presents foraminiferal and palynological results of the upper Badenian–Sarmatian strata in one borehole section in Poland, namely, Babczyn 2, located in the northeastern part of the Carpathian Foredeep, which is the largest Central Paratethyan Basin (Figure 1). The aim of this study was to identify environmental changes around the inferred Badenian/Sarmatian boundary. The studied borehole is a key section of the northern marginal part of the Carpathian Foredeep [3,6–8]. A recent biostratigraphic

study of two boreholes, including Babczyn 2 [8], confirmed, in general, zonation based on foraminifera that was previously established within the Badenian and Sarmatian strata of the northern margin of the Carpathian Foredeep [9–13] (cf. [14]; Figure 1). In the upper Badenian, the lower *Neobulimina longa* Zone and the upper *Cibicides crassiseptatus (Hanzawaia crassiseptata)* Zone occur (Figure 1); the latter is considered to be synchronous with the *Velapertina indigena* (Łuczkowska) planktonic foraminiferal zone [15].

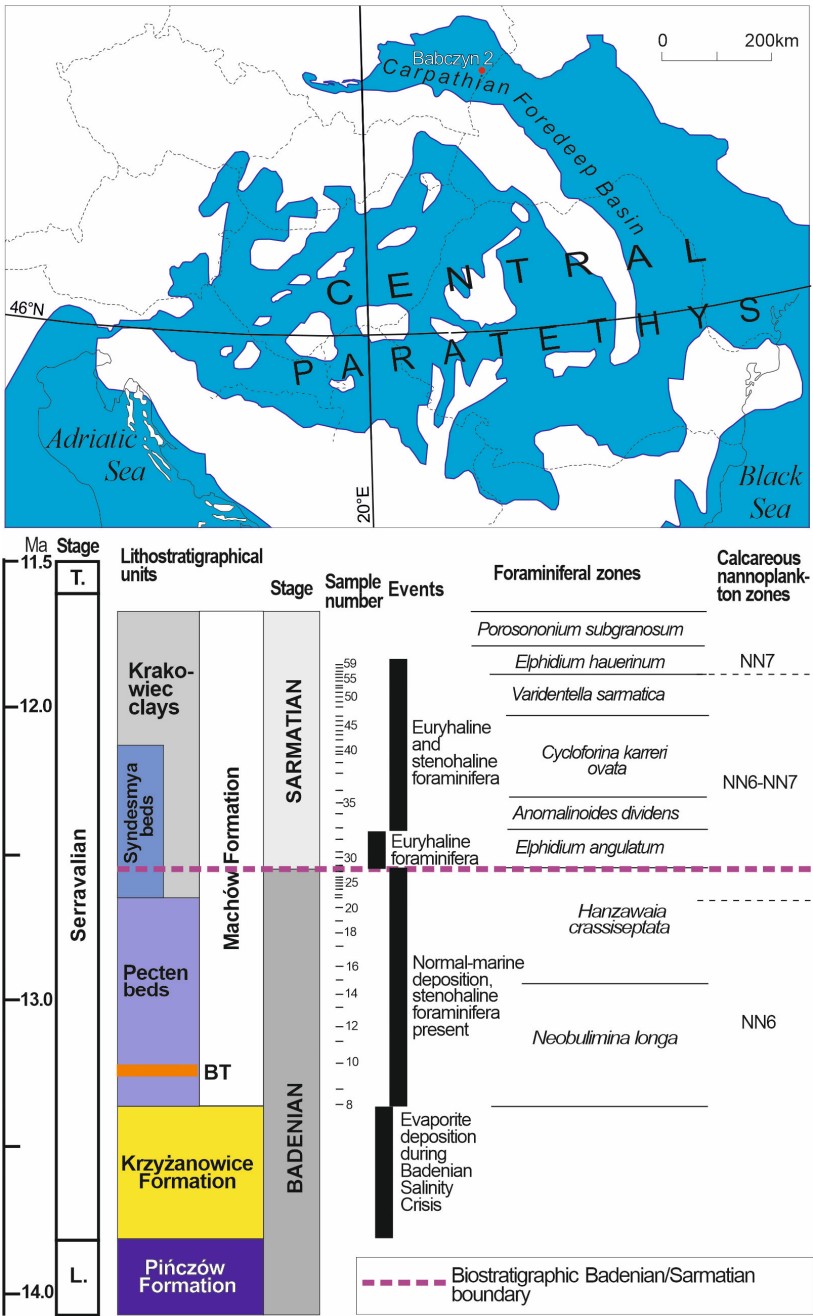

**Figure 1.** Central Paratethys realm in Middle Miocene (modified after [16]) and the location and stratigraphic position of the Babczyn 2 section (modified after [8]); BT—Babczyn Tuff (see [3]); L.—Langhian; T.—Tortonian; foraminiferal and calcareous nannoplankton zones after [8] except of *Porosononion subgranozum* zone recorded by [17].

Several foraminiferal zones have been identified in the Sarmatian strata (Figure 1). The lowest foraminiferal zone in the Babczyn 2 borehole was the *Elphidium angulatum* Partial

Range Zone occurring below the *Anomalinoides dividens* Interval Zone [8] that is commonly regarded as the lowest Sarmatian (eco)zone (e.g., [11,13,18,19]).

## 2. Geological Setting

The Carpathian Foredeep basin developed in front of the West Carpathians in the Early Miocene and is subdivided into inner and outer parts [20]. The fill is mostly composed of Middle Miocene (Badenian and Sarmatian) deposits. The Badenian and Sarmatian strata in this northern marginal part are several hundred metres thick, and towards the axial part of the Carpathian Foredeep, their thickness increases to more than 4 km.

The Langhian red-algal limestones and quartz arenites of the Pińczów Formation (Figure 1) originated from shelf and near-shore environments, and mudstones and clays originated from more basinal locations (e.g., [21]). Their thickness in the Babczyn 2 borehole is approximately 12 m. These are overlain by the Serravallian gypsum of the Krzyżanowice Formation [22] deposited during the Badenian Salinity Crisis ([23], with references therein). The gypsum sequence is 32 m thick [6]. In the uppermost part, an intercalation (2.3 m thick) of marly clays with marine palynomorphs (dinoflagellate cysts) and foraminiferal assemblages occurs [7]. It originated from a short-lived marine transgression to the Badenian evaporite basin prior to the transgression re-installing normal marine conditions in the Carpathian Foredeep Basin, which resulted from the reconnection of the basin with the Mediterranean and Eastern Paratethys primarily by tectonic modification of the interconnecting gateways [24].

Consequently, the Serravallian gypsum is overlain by the sandy–silty series of the Machów Formation (Figure 1). Its upper Badenian part is referred to as the Pecten Beds [25]. They are, in turn, overlain by Syndesmya Beds, which were considered to be Sarmatian by [25]. Ref. [3] confirmed the location of the Badenian–Sarmatian boundary between the Pecten and Syndesmya Beds at the highest occurrence of well-preserved and identifiable pectinid shells. The location was 6.0 m above the Babczyn Tuff found in the middle portion of the Pecten Beds and 3.4 m above the gypsum, which constrained its depositional age to $13.06 \pm 0.11$ Ma [3]. This age was calibrated against the same astronomically tuned FC standard (e.g., [24]), giving an age of $13.41 \pm 0.10$ Ma, which was comparable to the age of $13.32 \pm 0.07$ Ma recorded for a volcanic ash layer located several metres above the Badenian evaporites in the Romanian East Carpathians [24]. This recalibrated age was used as shown in Figure 1.

The Syndesmya Beds form the lowermost part of the Krakowiec Clays ([26,27], with references therein; Figure 1). Ref. [8] proved that the Badenian–Sarmatian boundary lies within the Syndesmya Beds (Figure 1) and not at their base as previously was assumed. The thickness of the Machów Formation in the Babczyn 2 borehole is almost 400 m; however, only the lowermost part of the formation was cored, and its coring started at a depth of 350 m [25].

## 3. Materials and Methods

A total of 51 samples from the 40.7 m thick interval (368.4–409.1 m deep) in the Babczyn 2 section were studied for foraminifera (samples 8–59); the same sample set was previously [8] used for a biostratigraphic study [8]. The sampling distance was ~1 m in the Pecten beds (samples 8–21), 0.5 m in the lowermost part of the Krakowiec clays (samples 22–29), and ~1 m in the higher part of the studied interval of the Krakowiec clays (samples 32–59). The most common and characteristic species are illustrated in Figures 2 and 3, respectively. The specimens in the figures were deposited at the Institute of Paleobiology, Polish Academy of Sciences in Warszawa, Poland (ZPAL F. 75).

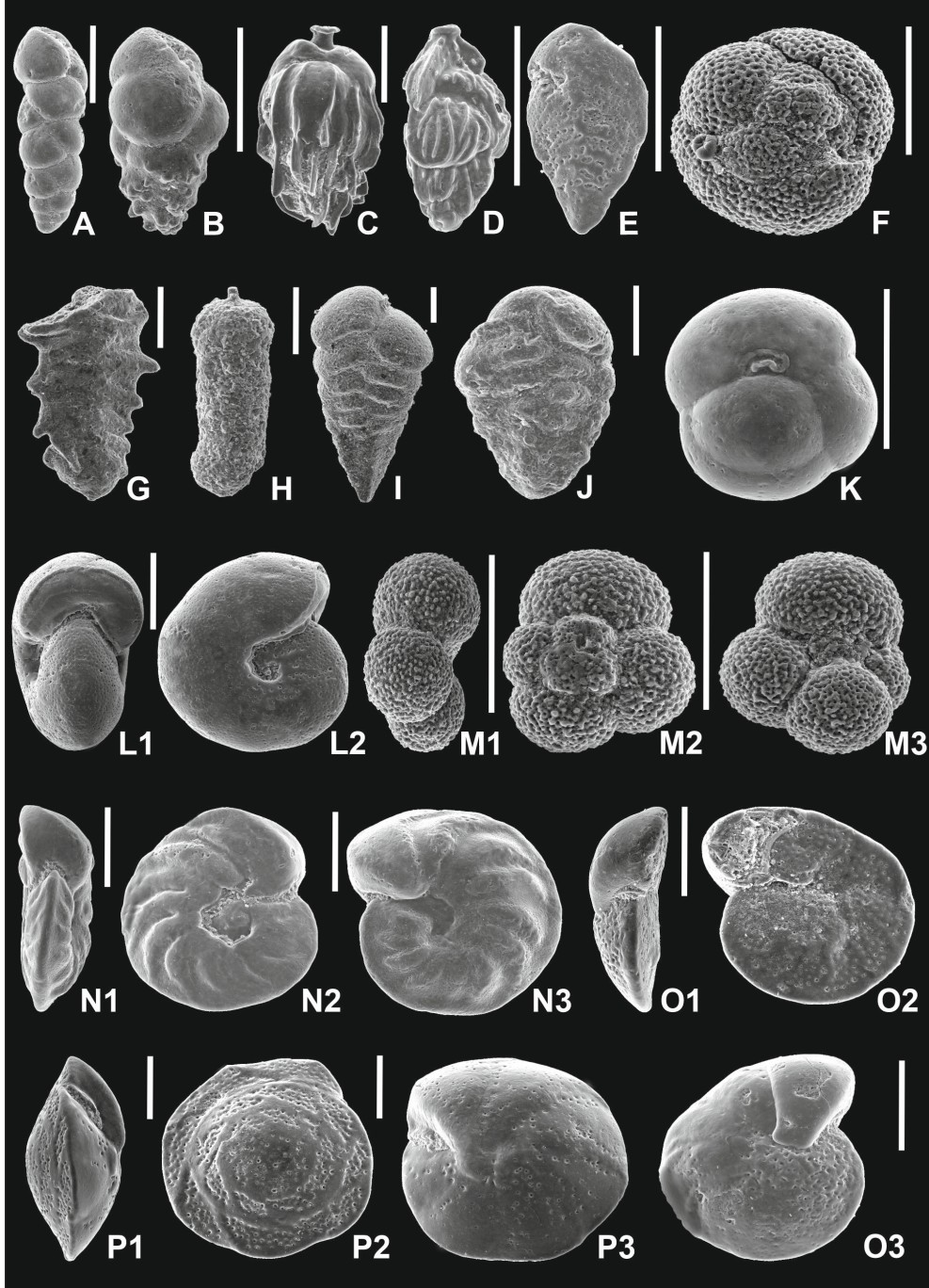

**Figure 2.** Foraminifera from the Babczyn 2 borehole. (**A**)—*Neobulimina longa* (Venglinskyi), sample 17; (**B**)—*Bulimina subulata* (Cushman and Parker), sample 17; (**C**)—*Uvigerina bellicostata* (Łuczkowska), sample 19; (**D**)—*Angulogerina angulosa* (Williamson), sample 20; (**E**)—*Bolivina dilatata* (Reuss), sample 17; (**F**)—*Velapertina indigena* (Łuczkowska), sample 28; (**G**)—*Spirorutilus carinatus* (d'Orbigny), sample 31; (**H**)—*Martinottiella communis* (d'Orbigny), sample 28; (**I**)—*Pseudogaudryina karreriana* (Cushman), sample 25; (**J**)—*Siphotextularia inopinata* (Łuczkowska), sample 28; (**K**)—*Sphaeroidina bulloides* (d'Orbigny), sample 11; (**L**)—*Melonis pompilioides* (Fichtel and Moll) [**L1**: apertural view; **L2**:lateral view], sample 22; (**M**)—*Globigerina bulloides* (d'Orbigny) [**M1**: edge view; **M2**: dorsal view; **M3**: ventral view], sample 28; (**N**)—*Hanzawaia crassiseptata* (Łuczkowska) [**N1**: edge view; **N2**: dorsal view; **N3**: ventral view], sample 28; (**O**)—*Lobatula lobatula* (Walker and Jacob) [**O1**: edge view; **O2**: dorsal view; **O3**: ventral view], sample 28; (**P**)—*Heterolepa dutemplei* (d'Orbigny) [**P1**: edge view; **P2**: dorsal view; **P3**: ventral view],, sample 28; scale bars—200 μm.

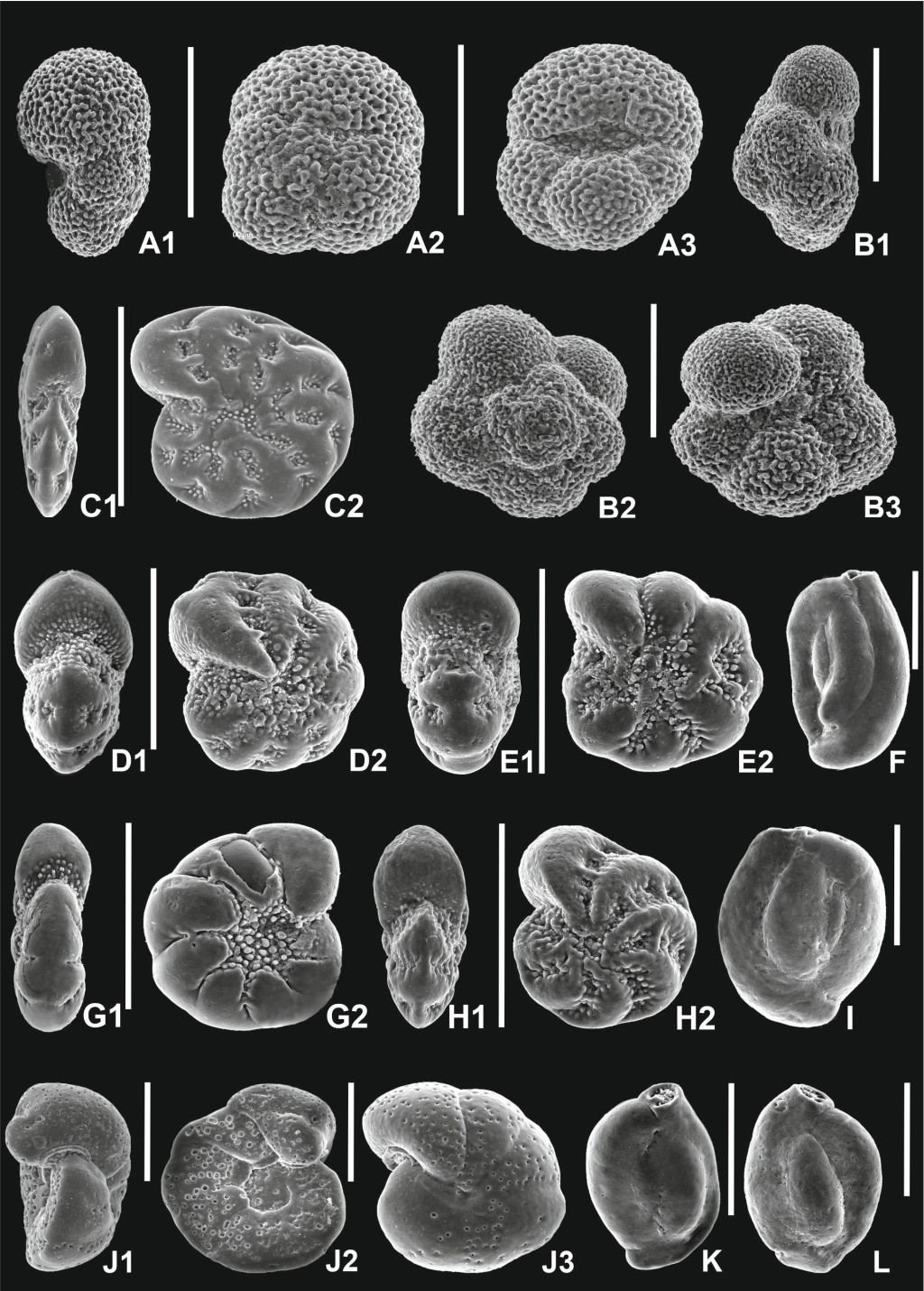

**Figure 3.** Foraminifera from the Babczyn 2 borehole. (**A**)—*Trilobatus* sp. [**A1**; edge view; **A2**: dorsal view; A3: ventral view],, sample 28; (**B**)—*Globigerina tarchanensis* (Subbotina and Chutzieva) [**B1**: edge view; **B2**: dorsal view; **B3**: ventral view], sample 28; (**C**)—*Elphidium advenum* subsp. *limbatum* (Chapman) [**C1**: apertural view; **C2**: lateral view], sample 33; (**D**)—*Elphidium angulatum* (Egger) [**D1**: apertural view; **D2**: lateral view], sample 29; (**E**)—*Elphidiella serena* (Venglinski) [**E1**: apertural view; **E2**: lateral view], sample 29; (**F**)—*Quinqueloculina akneriana* (d'Orbigny), sample 32; (**G**)—*Porosononion martkobi* (Bogdanovich) [**G1**: apertural view; **G2**: lateral view], sample 31; (**H**)—*Elphidium angulatum* (Egger) [**H1**: apertural view; **H2**: lateral view], sample 29; (**I,K,L**)—*Varidentella reussi* (Bogdanovich), sample 32; (**J**)—*Anomalinoides dividens* (Łuczkowska) [**J1**: edge view; **J2**: dorsal view; **J3**: ventral view], sample 35; scale bars—200 μm.

Depending on the foraminiferal abundances in different samples, all the foraminifera (up to 200 specimens) were picked. The relative abundances of planktonic and benthic foraminifera within the foraminiferal assemblages (P/B ratio), the Shannon–Wiener diversity indices H(S), the relative abundances of the most common genera, and the relative abundances of the infaunal and epifaunal morphogroups within the benthic foraminiferal assemblages were calculated [28,29] (Figure 4). H(S) values > 2.1 indicate normal marine environments [30]. The palaeoenvironmental interpretation based on the foraminifera involves the requirements of the present-day representatives of the recorded taxa [28–48].

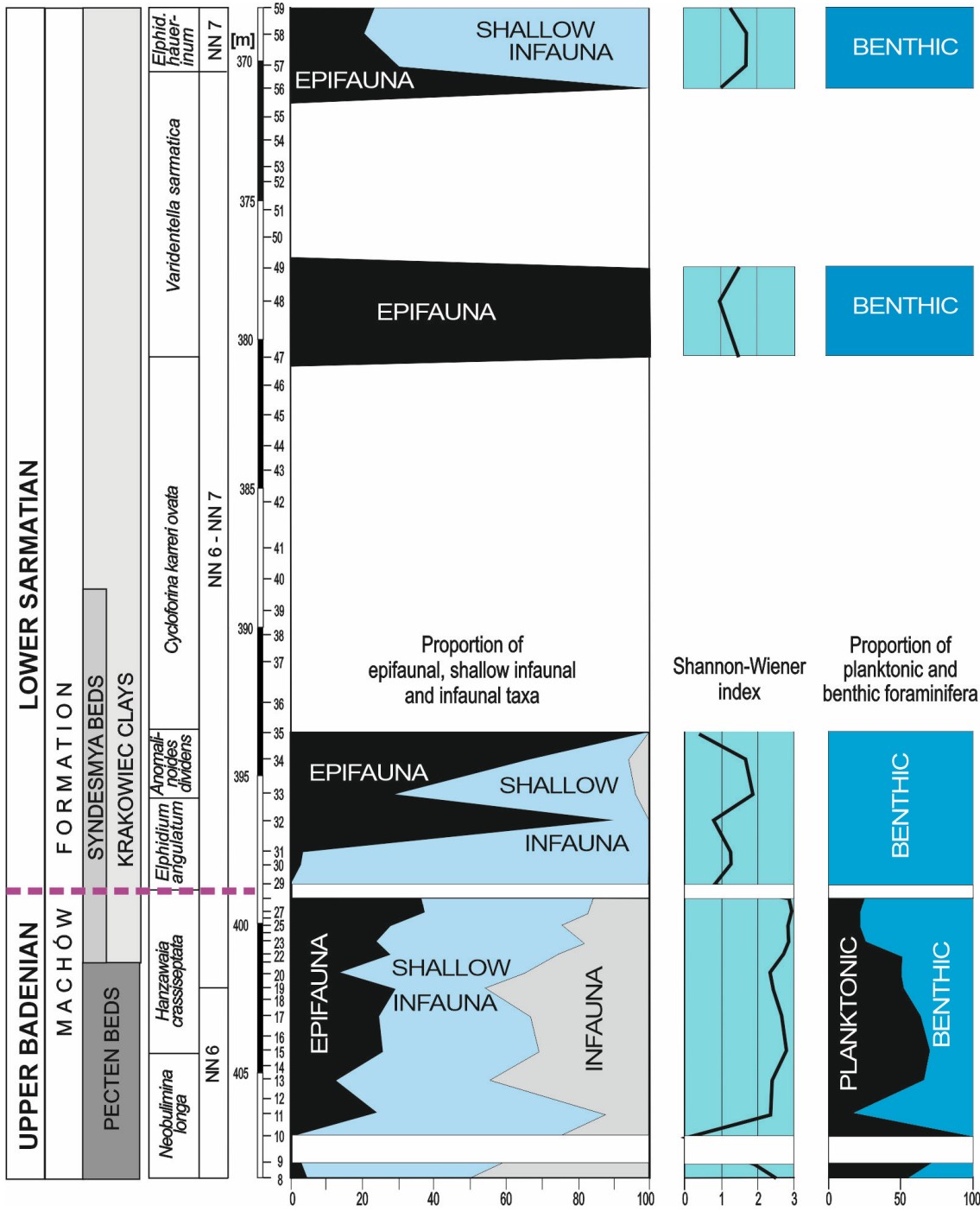

**Figure 4.** The relative abundances (in percentages) of epifaunal, shallow infaunal, and infaunal taxa, the Shannon–Wiener indices H(S) and percentages of planktonic foraminifera in foraminiferal assemblages (P/B ratios); purple dashed line indicates the Badenian/Sarmatian boundary.

The following taxa were included in the oxic group: *Anomalinoides* spp., *Cibicidoides* spp., *Hanzawaia* spp., *Heterolepa dutemplei*, *Lobatula lobatula*, *Valvulineria complanata*, *Globocassidulina subglobosa*, *Spirorutilus carinatus*, keeled elphidiids, and miliolids [39,43]. Oxic indices represent epifaunal species. Taxa tolerant of suboxic environments included *Astrononion perfossum*, *Hansenisca soldanii*, *Elphidiella serena*, *Melonis pompilioides*, *Pullenia bulloides*, *Valvulineria complanata*, *Angulogerina angulosa*, *Uvigerina* spp., and *Sphaeroidina bulloides* [39,43]; taxa tolerant of dysoxic environments included *Bolivina* spp. and *Bulimina* spp. Foraminifera tolerant of suboxic environments represent mostly shallow infaunal species, whereas foraminifera tolerant of dysoxic environments represent mostly deep infauna and species with opportunistic behaviours. These are commonly used as stress markers (e.g., [49–52]).

For palaeoecological analysis, species or groups of species that exceeded 5% in at least one sample were taken into account. There is a direct relationship between the abundance of species within a community and the environment. Abundance fluctuations of benthic foraminifera are sensitive palaeoceanographic indicators responding to changing palaeotemperature, salinity, nutrient supply, and oxygen conditions.

For palynology, 26 samples were studied (i.e., the odd sample numbers from 9 to 55 and the even numbers 8 and 28). The applied palynological procedure included 38% hydrochloric acid (HCl) treatment, 40% hydrofluoric acid (HF) treatment, heavy-liquid ($ZnCl_2$ + HCl; density 2.0 g·cm$^{-3}$) separation, ultrasonication for 10–15 s, and sieving at 10 μm on nylon mesh. No nitric acid ($HNO_3$) treatment was applied. Each processed rock sample weighed 10 g. Two slides from each sample were prepared using glycerine jelly as a mounting medium. The same set of slides was used for the analyses of dinoflagellate cysts, palynofacies (P.G.), and sporomorphs (E.W. and G.W.). Unprocessed rock samples, palynological residues, and slides were stored in the collection of the Institute of Geological Sciences at the Polish Academy of Sciences Research Centre in Kraków, Poland. The most common and characteristic species are illustrated in Figures 5 and 6, respectively.

A Zeiss Axiolab microscope equipped with 20× lens and 100× oil lens was used for the analyses of the palynofacies and dinoflagellate cysts. The quantitative dinoflagellate cyst analysis was conducted by counting about 300 specimens; in lower-frequency cases, all the specimens from both prepared slides were counted. On this basis, the Shannon–Wiener diversity index (H′) was calculated as follows: $H' = -\sum p_i \ln(p_i)$, where $p_i$ is the relative abundance of each taxon and is expressed as $e^{H'}$. The most frequent and important taxa for palaeoenvironmental reconstructions were grouped into eight dinoflagellate cyst morphogroups as follows:

- *Spiniferites* morphogroup, including all the *Spiniferites* and *Achomosphaera* specimens;
- *Batiacasphaera* morphogroup, including *B. sphaerica*, *Batiacasphaera* sp. A, *B. micropapillata*, and *B. hirsuta*;
- *Operculodinium* morphogroup, including all the *Operculodinium* specimens;
- *Systematophora* morphogroup, including *S. placacantha* and *S. ?ancyrea* specimens. The latter differed from *S. placacantha* because of incomplete or absent proximal ridges;
- *Lingulodinium* morphogroup, including *Lingulodinium machaerophorum* and a single specimen of *Lingulodinium* sp. A;
- *Nematosphaeropsis* morphogroup, including *N. labyrinthus*;
- *Impagidinium* morphogroup, including *Impagidinium* sp. and *I. strialatum* specimens;
- *Polysphaeridium* morphogroup, including *P. subtile* and *P. zoharyi*.

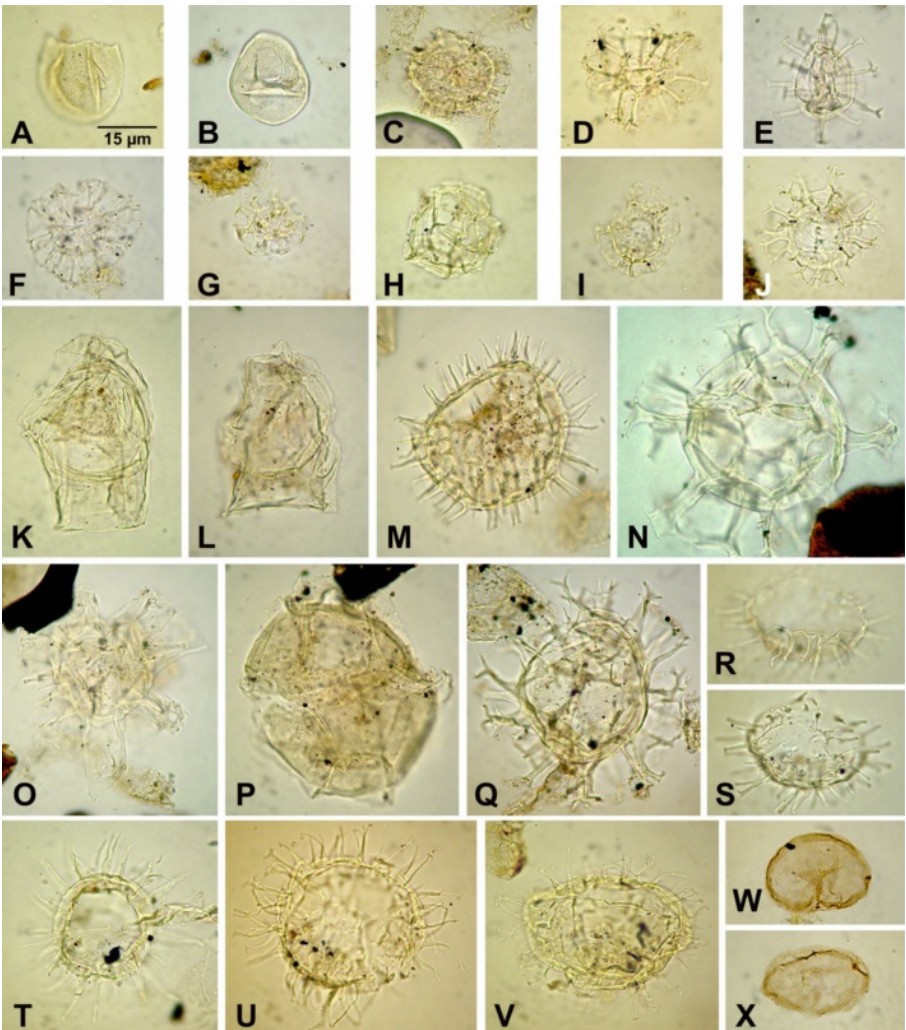

**Figure 5.** Dinoflagellate cysts from the Babczyn 2 borehole. (**A**)—*Batiacasphaera* sphaerica, sample 8; (**B**)—*Pyxidinopsis psilata*, sample 8; (**C**)—*Labyrinthodinium truncatum*, sample 9; (**D**)—*Reticulatosphaera actinocoronata*, sample 9; (**E**)—*Melitasphaeridium pseudorecurvatum*, sample 11; (**F**)—*Nematosphaeropsis labyrinthus*, sample 11; (**G**)—*Cordosphaeridium minimum*, sample 15; (**H**)—*Impagidinium* sp., sample 19; (**I**)—*Impagidinium* sp., sample 23; (**J**)—*Spiniferites ramosus* s.l., sample 23; (**K**)—*Hystrichosphaeropsis obscura*, sample 11; (**L**)—*Hystrichosphaeropsis obscura*, sample 11; (**M**)—*Operculodinium centrocarpum*, sample 11; (**N**)—*Spiniferites pseudofurcatus*, sample 15; (**O**)—*Hystrichokolpoma rigaudiae*, sample 15; (**P**)—*Pentadinium laticinctum*, sample 11; (**Q**)—*Spiniferites ramosus* s.l., sample 21; (**R,S**)—*Polysphaeridium subtile*, same specimen, various foci, sample 27; (**T**)—*Lingulodinium machaerophorum*, sample 23; (**U**)—*Systematophora placacantha*, sample 27; (**V**)—*Systematophora ?ancyrea*, sample 41; (**W**)—*Selenopemphix nephroides*, sample 41; (**X**)—*Selenopemphix nephroides*, sample 41. Scale bar in (**A**) refers to all the photomicrographs.

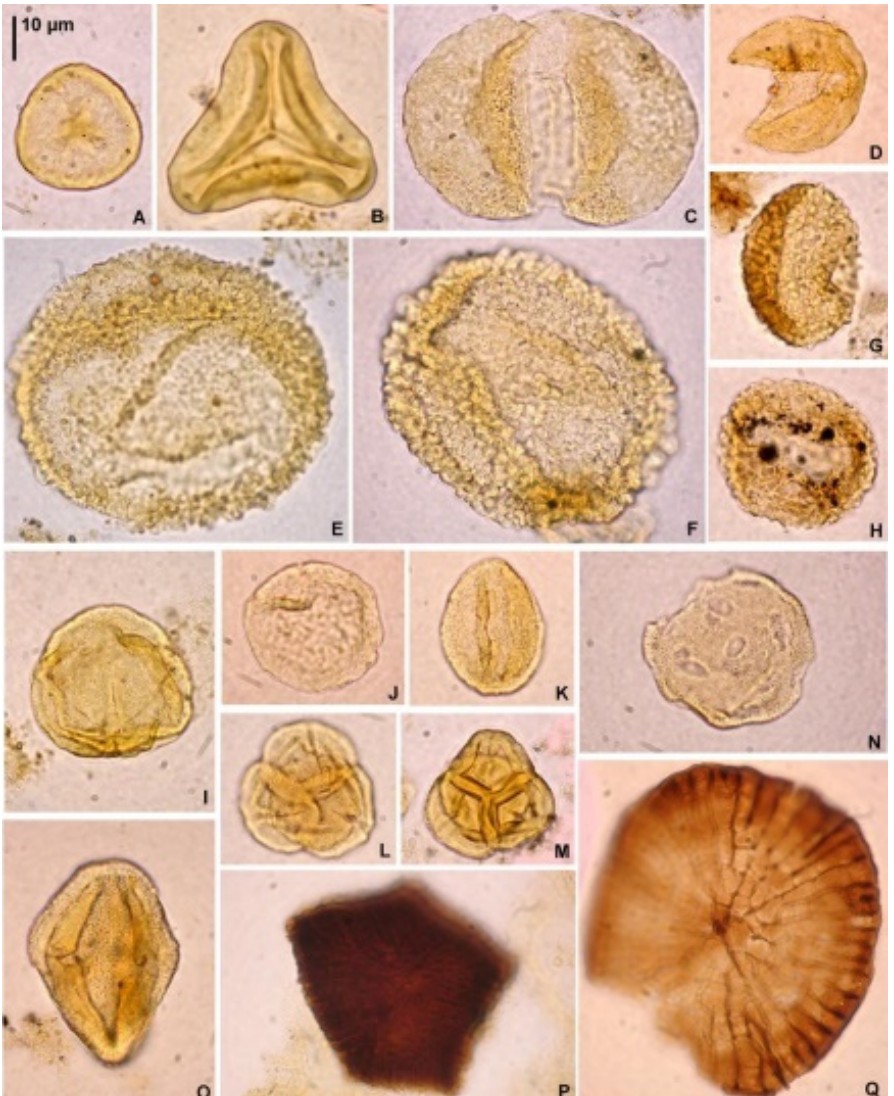

**Figure 6.** Terrestrial palynomorphs from the Babczyn 2 borehole. Spores of plants: (**A**)—*Stereisporites* sp., sample 55; (**B**)—*Neogenisporis* sp., sample 55. Pollen grains: (**C**)—*Cathayapollis* sp., sample 47; (**D**)—*Inaperturopollenites* cf. *verrupapillatus* Trevisan, sample 47; (**E**)—*Zonalapollenites gracilis* Krutzsch, sample 47; (**F**)—*Zonalapollenites verrucatus* Krutzsch, sample 55; (**G**)—*Sciadopityspollenites verticillatiformis* (Zauer) Krutzsch, sample 15; (**H**)—*Sciadopityspollenites verticillatiformis* (Zauer) Krutzsch, sample 33; (**I**)—*Faguspollenites* sp., sample 43; (**J**)—*Ulmipollenites undulosus* Wolff, sample 47; (**K**)—*Quercopollenites* sp., sample 55; (**L**)—*Ericipites baculatus* Nagy, sample 33; (**M**)—*Ericipites* sp., sample 55; (**N**)—*Periporopollenites orientaliformis* (Nagy) Kohlman-Adamska and Ziembińska-Tworzydło, sample 17; (**O**)—*Edmundipollis* sp., sample 9. Fungi: (**P**)—*Cephalothecoidomyces* cf. *neogenicus* G. Worobiec, Neumann and E. Worobiec, sample 47; (**Q**)—*Phragmothyrites* sp., sample 15. Scale bar in (**A**) refers to all the photographs.

The palynofacies analysis was performed by counting 500 particles. The palynofacies elements in the present study were assigned to nine groups as follows:

- Black, opaque phytoclasts representing the most resistant coalified land plant remains;
- Dark-brown phytoclasts, which are land plant remains, usually equidimensional, translucent (or with translucent edges), remaining in structureless wood or cortical tissues;
- Cuticle group, including structured land plant tissues and remains with usually preserved cell structures. These are commonly the remains of the leaf epidermis;

- Pollen grains of angiosperms and gymnosperms. The most common are bisaccate pollen grains of gymnosperms;
- Pteridophyte spores;
- Dinoflagellate cysts;
- Algae and the acritarch group, including algae other than dinoflagellate cysts, such as prasinophyte (e.g., *Tasmanites*, *Cymatiosphaera*, and *Leiosphaera*), and *incertae sedis* microfossils (i.e., acritarchs);
- Foraminifera organic linings (zooclasts), which are the organic cells of foraminifera;
- Amorphous organic matter (AOM), which represents structureless particles of uncertain origins (most likely marine origin in the case of the present study materials), representing a stage in the bacterial decay of organic particles in oxygen-depleted environments.

The identified sporomorph taxa were classified based on the *Atlas of Pollen and Spores of the Polish Neogene* [53–56]. In the studied material, the following palaeofloristical elements were distinguished: "palaeotropical" (P), including "tropical" (P1) and "subtropical" (P2); "arctotertiary" (A), including "warm-temperate" (A1) and "temperate" (A2); and cosmopolitan (P/A). The data from the palynological spectra were used to construct a palynological diagram. In the diagram, the percentage shares of the pollen and spore taxa were calculated from the total sum of the pollen grains and spores, and the proportion of non-pollen palynomorphs was computed separately in relation to the total sum using POLPAL software [57].

Microphotographs of the selected sporomorphs and fungal palynomorphs (Figure 6) were obtained using a Nikon Eclipse E400 microscope equipped with a Canon A640 digital camera.

## 4. Results

### 4.1. Foraminifera

The Babczyn 2 succession contains well-preserved planktonic, calcareous, and agglutinated benthic foraminifera (Appendix 1 in [8]). Altogether, 60 species, representing the following genera were observed: *ArtiSculina*, *Cycloforina*, *Pseudotriloculina*, *Quinqueloculina*, *Sigmoilinita*, *Triloculina*, *Varidentella*, *Elphidium*, *Haynesina*, *Porosononion*, *Anomalinoides*, *Heterolepa*, *Hanzawaia*, *Lobatula*, *Cibicidoides*, *Elphidiella*, *Melonis*, *Nonion*, *Pullenia*, *Bolivina*, *Bulimina*, *Uvigerina*, *Angulogerina*, *Sphaeroidina*, *Favulina*, *Globigerina*, *Velapertina*, *Trilobatus*, *Ammobaculites*, *Ammodiscus*, *Martinottiella*, *Siphotextularia*, *Textularia*, *Spirorutilus*, *Pavonitina*, *Pseudogaudryina*, *Rhabdammina*, and *Nothia*.

In the Badenian part of the studied succession comprising the Pecten Beds and lowermost part of the Syndesmya Beds, both planktonic and benthic foraminifera occur. However, in sample 9, the tests of the foraminifera are damaged in most cases, indicating their transportation and reworking. Sample 10 possesses almost entirely planktonic foraminifera; rare specimens of benthic foraminifera were recorded. Samples from 11 to 28 yielded abundant and well-preserved foraminiferal assemblages, and sample 29 is almost devoid of foraminifera. Upsection, the samples from 30 to 35 from the lowermost Sarmatian yielded only benthic foraminifera, and in the upper part of the section (samples from 35 to 55), foraminifera are rare, and their assemblage is composed of miliolids and reworked forms from the Badenian. Only in a short interval (samples from 47 to 49), the abundant assemblage of miliolids occurs, and in samples 50 and 51, *Bolivina sarmatica* with miliolids is recorded. Owing to small numbers of specimens, quantitative analysis was not performed for the upper part of the studied section.

The following genera exceed 5% in at least one sample: *Bulimina*, *Bolivina*, *Uvigerina*, *Angulogerina*, *Cibicidoides*, *Melonis*, *Sphaeroidina*, *Hanzawaia*, *Spirorutilus*, *Heterolepa*, *Elphidium*, and *Porosononion* (Figure 7). Quantitative analysis enabled the grouping of samples with homogeneous foraminiferal assemblages. Nine foraminiferal assemblages were recognised in the studied interval (Figure 7).

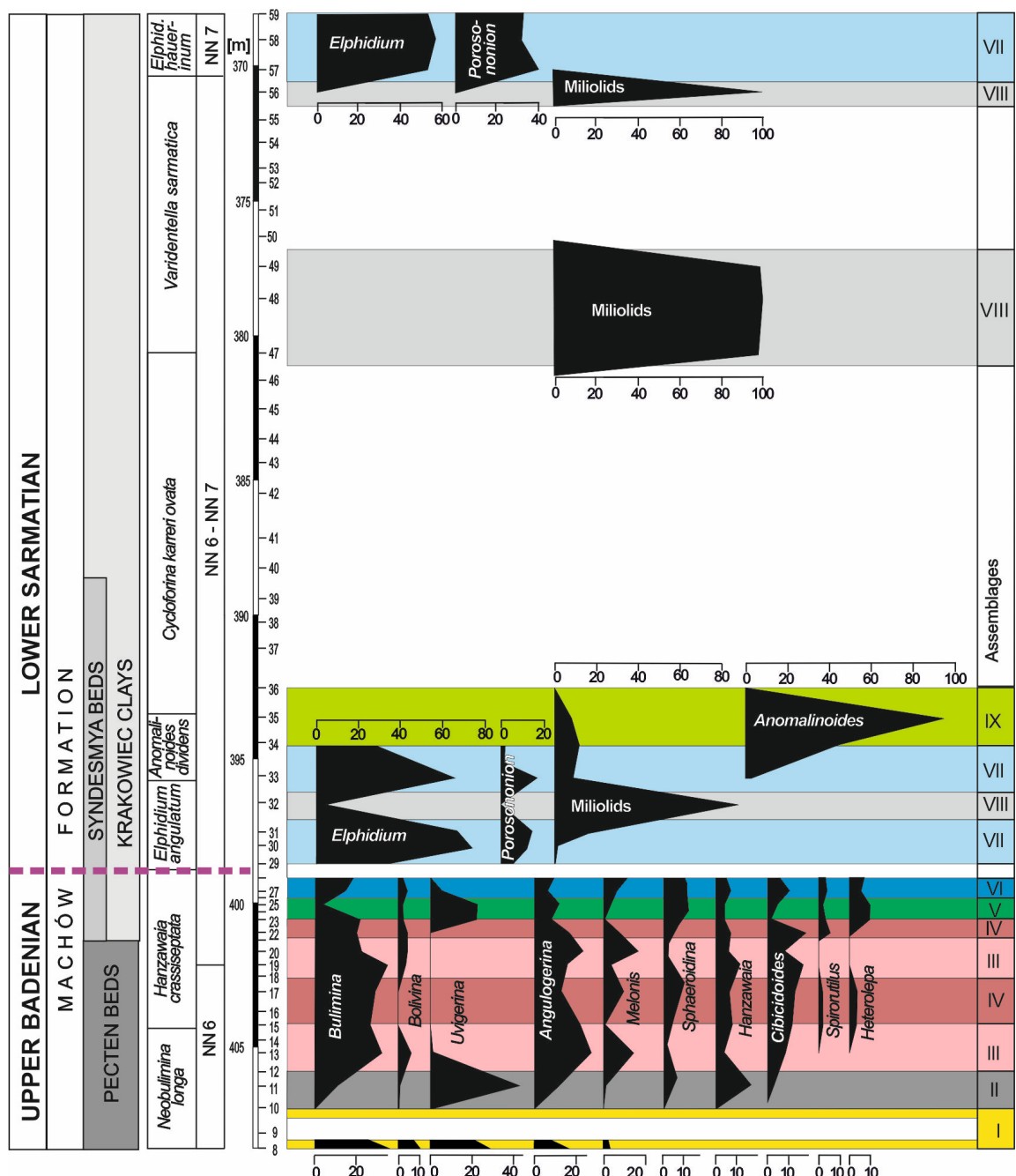

**Figure 7.** Relative abundances of the most common benthic foraminiferal genera (in percentages) of the Babczyn 2 borehole; purple dashed line indicates the Badenian/Sarmatian boundary.

Assemblage I occurs at the base of the section (sample 8); sample 9, because of large numbers of resedimented specimens, has not been taken into account (Figure 7). This assemblage is characterised by the dominance of *Bulimina* (30%) and *Uvigerina* (30%). Other important components are *Angulogerina*, which exceeds 17%, and *Bolivina* (10%). Subsidiary species include *Melonis pompilioides*, *Globocassidulina subglobosa*, *Astrononion perfossum*, *Asterigerinata planorbis*, and *Lobatula lobatula*. The P/B ratio exceeds 60%; the H(S) diversity index is 2.5. Infaunal taxa comprise 95% of the assemblage.

Assemblage II is recorded in samples 10 and 11 (Figure 7). It is dominated by *Uvigerina*, which reaches 40% in the benthic foraminiferal assemblage, and contains *Bulimina* and *Angulogerina*, which exceed 10% (18 and 12%, respectively). *Hanzawaia* (16%) and

*Cibicidoides* (11%) are other common components of this assemblage. The P/B ratio is 17%; the H(S) diversity index ranges from 2.3 to 2.5.

Assemblage III occurs in samples 13, 19, and 20 (Figure 7). It is dominated by *Bulimina* (from 30 to 40%), *Angulogerina* (from 16 to 28%), and *Melonis* (from 14 to 17%). Minor components of the assemblage are *Bolivina*, *Cibicidoides*, *Sphaeroidina*, and *Hanzawaia*. P/B varies from 70 to 80%; H(S) rangers between 2.3 and 2.7; infaunal species comprise from 75 to 85% of the assemblage (Figure 4).

Assemblage IV is recorded in samples 15, 17, and 22 (Figure 7). The dominant and common species are *Bulimina* (from 20 to 34%), *Angulogerina* (from 12 to 17%), *Sphaeroidina* (from 7 to 10%), *Cibicidoides* (13%), and *Hanzawaia* (8%). In this assemblage, for the first time in the studied interval, the following agglutinated foraminifera appear: *Spirorutilus*, *Siphotextularia*, and *Ammodiscus*. P/B is high and reaches 65%; H(S) is from 2.6 to 2.7; infaunal species comprise from 70 to 75% of the assemblage (Figure 4).

Assemblages III and IV alternate twice in the middle and upper parts of the Pecten Beds. They differ from the under- and overlying assemblages by a complete lack of *Uvigerina.* Instead, *Angulogerina angulosa* is one of dominant taxa and forms from 12% to 28% of these assemblages.

Assemblage V occurs in samples 23 and 25 (Figure 7). In these samples, *Uvigerina* reappears, and it forms >20% of the assemblage. The contents of three other species are around 10%: *Heterolepa dutemplei* (10%), *Angulogerina angulosa* (8–10%), and *Sphaeroidina bulloides* (12%). P/B is about 25%, the H(S) diversity index is 2.8, and infaunal morphogroups form 80% of the assemblage (Figure 4).

Assemblage VI is recorded in samples 27 and 28 (Figure 7). *Bulimina* forms from 15 to 19%, while *Uvigerina* drops to 0–6%. *Angulogerina angulosa* forms from 6 to 9%, *Sphaeroidina bulloides* forms from 8 to 10%, *Heterolepa dutemplei* forms from 5 to 7%, and agglutinated taxa (*Siphotextularia inopinata*, *Pseudogaudryina karreriana*, *Ammodiscus* sp., and *Spirorutilus carinatus*) form 13% of the assemblage. P/B is low, from 22 to 27%; H(S) is from 2.9 to 3.0; infaunal species comprise from 65 to 70% of the assemblage (Figure 4). This abundant, highly diversified foraminiferal assemblage suddenly disappeared at a depth of 399.4 m (sample 28).

Assemblage VII is recorded in samples 29, 30, 31, 33, 57, 58, and 59 (Figure 7). It is dominated by small *Elphidium*, which form from 60 to almost 80% of the assemblage; *Porosononion* and miliolids form from 10 to 20%. H(S) is from 1.8 to 1.9; infaunal species account for from 70 to 95% (Figure 4).

Assemblage VIII occurs in samples 32, 47, 48, 49, and 56 (Figure 7). It is very poorly diversified and almost completely comprises a miliolid assemblage. H(S) is 0.77; only epifauna occur (Figure 4).

Assemblage IX is recorded in samples 34 and 35 (Figure 7). It is an almost monospecific assemblage of *Anomalinoides dividens* (from 50 to 90%). Miliolids form from 8 to 12%, and H(S) is from 0.33 to 1.2 (Figure 4).

*4.2. Palynofacies*

All the samples yielded palynological organic matter, the elements of which showed various ratios reflecting variable environmental conditions and/or various sedimentological processes (Figure 8). The palynofacies of the studied samples were dominated by terrestrial elements represented by palynodebris (black and dark-brown phytoclasts and cuticles) and pollen grains. The shares of these two groups showed variable proportions: palynodebris (cuticles in particular) dominated in the lower part of the studied interval and decreased moving upwards, being replaced by pollen grains in the middle part and AOM in the upper part (Figure 8).

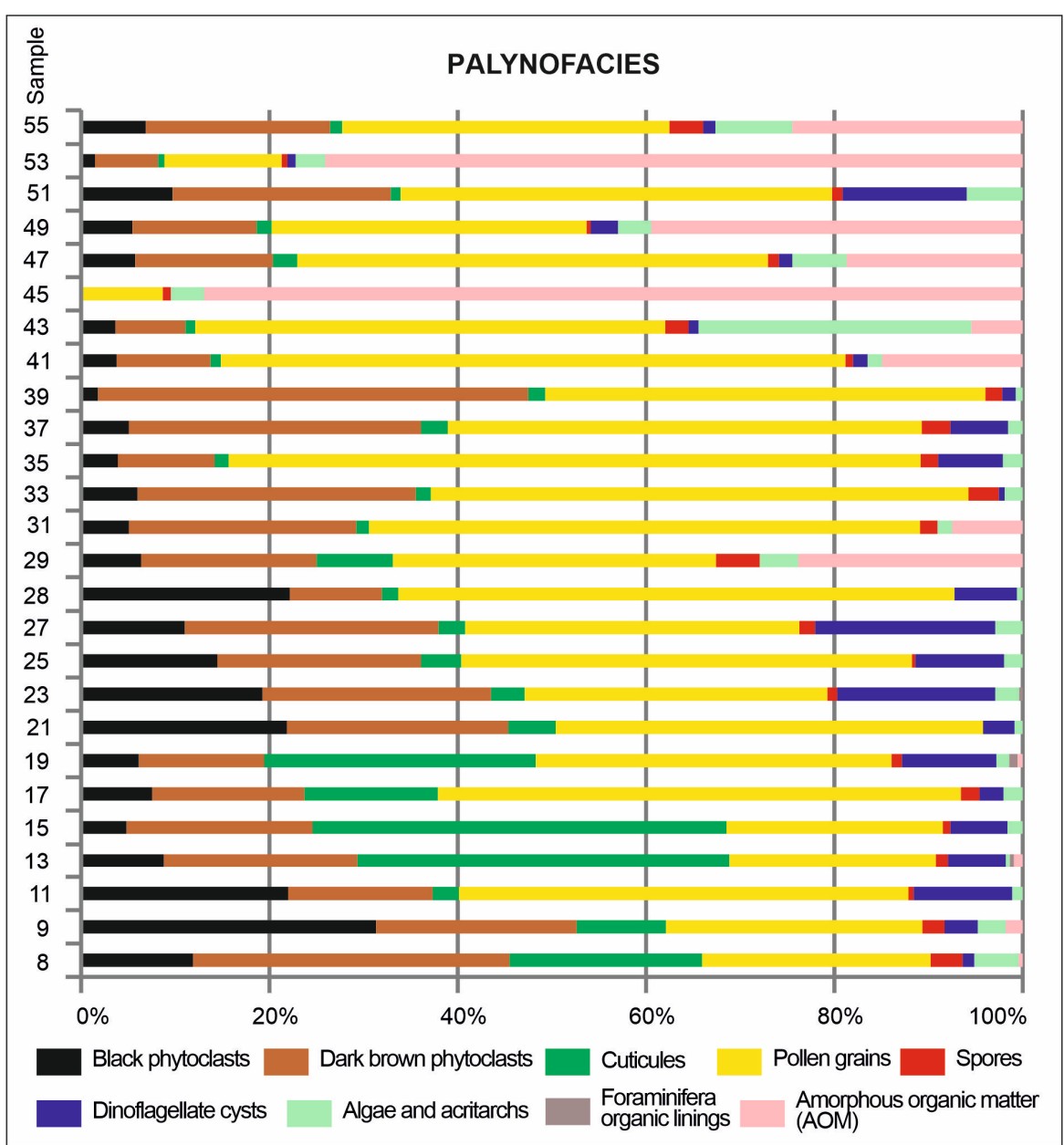

**Figure 8.** Palynofacies changes in the Babczyn 2 borehole.

The proportion of marine elements (dinoflagellate cysts—Figure 5, prasinophytes, acritarchs, and rare zooclasts) rarely exceeded 10%. Dinoflagellate cysts were the most common in the lower and middle intervals while becoming rare or absent in the upper part, where prasinophytes and acritarchs predominated among the aquatic palynomorphs (Figure 8).

AOM occurred in two samples from the middle part of the studied interval (samples 29 and 31) and in the upper part, where it dominated in some samples (Figure 8).

### 4.3. Dinoflagellate Cysts

Virtually all the studied samples (except for samples 29, 31, and 45) involved dinoflagellate cysts. A total of 44–48 taxa assumed to be in situ were distinguished. We note that some taxa, such as *Spiniferites ramosus* s.l., included several species and/or subspecies, which were not distinguished for the purposes of the present study (Tables 1–4). Despite this apparent taxonomical richness, the taxonomic diversity of the dinoflagellate cyst as-

semblages was low, and it oscillated between 2.0 and 6.5 (Figure 9). The majority of the samples yielded assemblages dominated by from three to four species, with the remaining taxa being represented by rare or even single specimens. The dominant taxa included *Spiniferites ramosus* s.l., *Batiacasphaera* spp., and *Systematophora* spp., with the most common ones being *Operculodinium* spp. and *Lingulodinium machaerophorum*; their proportions varied from sample to sample and could also be different in neighbouring samples. Some species, like *Nematosphaeropsis labyrinthus*, *Selenopemphix nephroides*, *Reticulatosphaera actinocoronata*, *Lingulodinium machaerophorum*, and *Polysphaeridium subtile*, showed increased frequencies only in single samples.

**Table 1.** Distribution of aquatic palynomorphs in the upper Badenian of Babczyn 2 borehole (part 1). (*) indicates reworked taxa; their occurrence is not included in the total count.

| Age | Upper Badenian | | | | | | | | | | | |
|---|---|---|---|---|---|---|---|---|---|---|---|---|
| **Lithostratigraphy** | **Machów Formation** | | | | | | | | | | | |
| | **Pecten Beds** | | | | | | | | **Syndesmya Beds** | | | |
| **Sample** | **8** | **9** | **11** | **13** | **15** | **17** | **19** | **21** | **23** | **25** | **27** | **28** |
| *Dinoflagellate Cysts* | | | | | | | | | | | | |
| 1. *Pentadinium laticinctum* | 13 | | 1 | 2 | 2 | | | | 1 | | 4 | 1 |
| 2. *Batiacasphaera sphaerica* | 68 | 3 | 3 | 4 | 5 | 3 | 1 | 20 | 22 | | 3 | 2 |
| 3. *Spiniferites ramosus s.l.* | 81 | 162 | 135 | 120 | 117 | 5 | 138 | 2 | 153 | 65 | 109 | 128 |
| 4. *Batiacasphaera* sp. A | 79 | 12 | 87 | 43 | 41 | 44 | 32 | 16 | 51 | | 1 | |
| 5. *Operculodinium centrocarpum* | 27 | 51 | 5 | 59 | 96 | 5 | 37 | 9 | 18 | 3 | 25 | 29 |
| 6. *Melitasphaeridium ?pseudorecurvatum* | 3 | | | | | | | | | | | |
| 7. *Homotryblium tenuispinosum* * | * | | | | * | | * | | | | | |
| 8. *Melitasphaeridium choanophorum* | 4 | | 15 | 3 | 13 | | 4 | | 6 | | 3 | 12 |
| 9. *Pyxidinopsis psilata* | 7 | | 3 | 1 | | 1 | | | 2 | | 1 | 12 |
| 10. *Systematophora placacantha* | 6 | 41 | 16 | 10 | 12 | | 5 | | 2 | 1 | 7 | 3 |
| 11. *Lingulodinium machaerophorum* | 9 | 5 | 11 | 14 | 2 | | 15 | | 12 | 3 | 5 | |
| 12. *Labyrinthodinium truncatum* | 5 | 10 | 12 | | 7 | | 2 | | | | 2 | 2 |
| 13. *Reticulatosphaera actinocoronata* | 1 | 3 | 5 | 1 | 6 | | 18 | | 6 | | 2 | |
| 14. *Dapsilidinium* sp. | 2 | | | | | | | | | | | |
| 15. *Operculodinium* sp. C | | 2 | | | | | | | | | | |
| 16. *Spiniferites pseudofurcatus* | | 1 | 2 | 1 | 3 | | 5 | | | | | |
| 17. *Hystrichostrogylon* sp. | | 1 | | | | | | | | | | |
| 18. *Reticulatosphaera?* sp. | | 1 | | | | | | | | | | |
| 19. *Spiniferites* sp. A | | 3 | 4 | 4 | | | 2 | | 2 | 3 | 3 | 6 |
| 20. *Lejeunecysta* sp. | | 1 | | | | | | | | 1 | 2 | |
| 21. *Hystrichosphaeropsis obscura* | | 1 | 4 | | | | | | | | | |
| 22. *Pyxidinopsis?* sp. | | 1 | | | 1 | | | | | | | |
| 23. *Cordosphaeridium* cf. *minimum* | | | 2 | | 1 | | | | 8 | | 1 | |
| 24. *Melitasphaeridium pseudorecurvatum* | | | 1 | 1 | 1 | 1 | 1 | | | | 4 | 1 |

**Table 1.** *Cont.*

| Age | Upper Badenian | | | | | | | | | | | |
|---|---|---|---|---|---|---|---|---|---|---|---|---|
| Lithostratigraphy | Machów Formation | | | | | | | | | | | |
| | Pecten Beds | | | | | | | Syndesmya Beds | | | | |
| Sample | 8 | 9 | 11 | 13 | 15 | 17 | 19 | 21 | 23 | 25 | 27 | 28 |
| 25. *Nematosphaeropsis labyrinthus* | | | 6 | | 1 | | | | 15 | 1 | 119 | |
| 26. *Cerodinium* sp. * | | | * | | | | | | | | | * |
| 27. *Homotryblium plectilum* * | | | * | | | | | | | | * | * |
| 28. *Impagidinium* sp. | | | 1 | | 1 | | 6 | | 5 | | 1 | |
| 29. *Hystrichokolpoma rigaudiae* | | | 2 | 1 | 1 | | | | | | | 2 |
| 30. *Cordosphaeridium minimum* | | | 1 | | 1 | | | | 1 | | 3 | |
| 31. *Operculodinium* sp. B | | | 4 | | 2 | | | | | | | |
| 32. *Wetzeliella* sp. * | | | * | | | | | | | | | * |
| 33. *Charlesdowniea* sp. * | | | * | | | | | | | | | |
| 34. *Operculodinium* sp. A | | | 1 | 2 | | 1 | | | | | | |
| 35. *Areosphaeridium* sp. * | | | | | * | | | | | | | |
| 36. *Glaphyrocysta semitecta* * | | | | | * | | | | | | | |
| 37. *Areosphaeridium diktyoplokum* * | | | | | * | | | | * | | * | |
| 38. *Deflandrea phosphoritica* * | | | | | | * | | | | | | * |
| 39. *Lingulodinium* sp. A | | | | | | | 1 | | | | | |
| 40. *Cleistosphaeridium* sp. | | | | | | | 3 | | | | | |
| 41. *Areoligera* sp. * | | | | | | | * | | | | | * |

**Table 2.** Distribution of aquatic palynomorphs in the lower Sarmatian of Babczyn 2 borehole (part 1). (*) indicates reworked taxa; their occurrence is not included in the total count.

| Age | Lower Sarmatian | | | | | | | | | | | | |
|---|---|---|---|---|---|---|---|---|---|---|---|---|---|
| Lithostratigraphy | Machów Formation | | | | | | | | | | | | |
| | Syndesma Beds | | | | | | | Krakowiec Clays | | | | | |
| Sample | 29 | 31 | 33 | 35 | 37 | 39 | 41 | 43 | 47 | 49 | 51 | 53 | 55 |
| *Dinoflagellate Cysts* | | | | | | | | | | | | | |
| 1. *Pentadinium laticinctum* | | | 3 | 1 | 2 | | | | | 13 | 1 | | |
| 2. *Batiacasphaera sphaerica* | | | | 9 | | | | | | 1 | | | |
| 3. *Spiniferites ramosus s.l.* | | | 12 | 242 | 135 | 3 | 25 | 24 | 52 | 141 | 12 | 31 | 50 |
| 4. *Batiacasphaera* sp. A | | | | 1 | | | | | | | 127 | | |
| 5. *Operculodinium centrocarpum* | | | | 8 | 10 | 2 | | | | 22 | 3 | | |
| 6. *Melitasphaeridium ?pseudorecurvatum* | | | | | | | | | | | | | |
| 7. *Homotryblium tenuispinosum* * | | | | | | | | | | | | | |
| 8. *Melitasphaeridium choanophorum* | | | | | 2 | | | | | | | | |
| 9. *Pyxidinopsis psilata* | | | | | | | | | | | | | |
| 10. *Systematophora placacantha* | | | 4 | | 24 | | 3 | | 71 | 72 | 3 | 78 | 31 |

**Table 2.** *Cont.*

| | Age | | | | | | Lower Sarmatian | | | | | | | |
|---|---|---|---|---|---|---|---|---|---|---|---|---|---|---|
| | Lithostratigraphy | | | | | | Machów Formation | | | | | | | |
| | | | | Syndesma Beds | | | | | | Krakowiec Clays | | | | |
| | Sample | 29 | 31 | 33 | 35 | 37 | 39 | 41 | 43 | 47 | 49 | 51 | 53 | 55 |
| 11. | *Lingulodinium machaerophorum* | | | | 13 | 15 | 27 | | | | | 13 | | |
| 12. | *Labyrinthodinium truncatum* | | | | 1 | | | | | | | | | |
| 13. | *Reticulatosphaera actinocoronata* | | | 1 | 25 | | | | | | | | | |
| 14. | *Dapsilidinium* sp. | | | | | | | | | | 2 | | | 1 |
| 15. | *Operculodinium* sp. C | | | | | | | | | | | | | |
| 16. | *Spiniferites pseudofurcatus* | | | | 1 | | | | | | | | | |
| 17. | *Hystrichostrogylon* sp. | | | | | | | | | | | | | |
| 18. | *Reticulatosphaera*? sp. | | | | | | | | | | | | | |
| 19. | *Spiniferites* sp. A | | | | | | | 6 | | | 1 | | | |
| 20. | *Lejeunecysta* sp. | | | | | | | | | | 3 | | 1 | |
| 21. | *Hystrichosphaeropsis obscura* | | | | | | | | | | | | | |
| 22. | *Pyxidinopsis*? sp. | | | | | | | | | | | | | |
| 23. | *Cordosphaeridium* cf. *minimum* | | | | 1 | | | | | 1 | 5 | | 12 | 9 |
| 24. | *Melitasphaeridium pseudorecurvatum* | | | | 1 | | | | | | | | | |
| 25. | *Nematosphaeropsis labyrinthus* | | | | | | | | | | | | | |
| 26. | *Cerodinium* sp. * | | | | | | | | | | | | | |
| 27. | *Homotryblium plectilum* * | | | | | | | | | | | | | |
| 28. | *Impagidinium* sp. | | | | | | | | | | 1 | | | |
| 29. | *Hystrichokolpoma rigaudiae* | | | | | | | | | | | | | |
| 30. | *Cordosphaeridium minimum* | | | | 2 | 3 | | | | | 6 | | | |
| 31. | *Operculodinium* sp. B | | | | | | | | | | | | | |
| 32. | *Wetzeliella* sp. * | | | | | | | | | | | | | |
| 33. | *Charlesdowniea* sp. * | | | | | | | | | | | | | |
| 34. | *Operculodinium* sp. A | | | | | | | | | | | | | |
| 35. | *Areosphaeridium* sp. * | | | | | | | | | | | | | |
| 36. | *Glaphyrocysta semitecta* * | | | | | | | | | | | | | |
| 37. | *Areosphaeridium diktyoplokum* * | | | | | | | | | | | | | |
| 38. | *Deflandrea phosphoritica* * | | | | | | | | | | | | | |
| 39. | *Lingulodinium* sp. A | | | | | | | | | | | | | |
| 40. | *Cleistosphaeridium* sp. | | | | | | | | | | | | | |
| 41. | *Areoligera* sp. * | | | | | | | | | | | | | |
| 42. | *Polysphaeridium subtile* | | | 16 | | 39 | | | | | | | | |

**Table 3.** Distribution of aquatic palynomorphs in the upper Badenian of Babczyn 2 borehole (part 2). (*) indicates reworked taxa; their occurrence is not included in the total count.

| | Age | Upper Badenian | | | | | | | | | | | |
|---|---|---|---|---|---|---|---|---|---|---|---|---|---|
| | Lithostratigraphy | Machów Formation | | | | | | | | | | | |
| | | Pecten Beds | | | | | | | | Syndesmya Beds | | | |
| | Sample | 8 | 9 | 11 | 13 | 15 | 17 | 19 | 21 | 23 | 25 | 27 | 28 |
| | *Dinoflagellate Cysts* | | | | | | | | | | | | |
| 42. | *Polysphaeridium subtile* | | | | | | | 19 | | 1 | 5 | 5 | |
| 43. | *Cordosphaeridium* sp. * | | | | | | | * | | | | | * |
| 44. | *Batiacasphaera micropapillata* | | | | | | | | 6 | | | | |
| 45. | *Polysphaeridium zoharyi* | | | | | | | | 1 | | | | 5 |
| 56. | *Batiacasphaera hirsuta* | | | | | | | | | 13 | 95 | 10 | 39 |
| 47. | *Membranophoridium aspinatum* * | | | | | | | | | * | | | |
| 48. | *Impagidinium pallidum* | | | | | | | | | 1 | 3 | 2 | |
| 49. | *Areosphaeridium michoudii* * | | | | | | | | | | * | | |
| 50. | *Operculodinium*? sp. | | | | | | | | | | 1 | 5 | 2 |
| 51. | *Palaeohystrichophora*? sp. * | | | | | | | | | | | * | |
| 52. | *Cleistosphaeridium* sp. A * | | | | | | | | | | | | * |
| 53. | *Operculodinium* sp. D | | | | | | | | | | | | 3 |
| 54. | *Operculodinium* sp. E | | | | | | | | | | | | 1 |
| 55. | *Operculodinium* sp. F | | | | | | | | | | | | 7 |
| 56. | *Hystrichokolpoma* sp. | | | | | | | | | | | | 3 |
| 57. | *Achomosphaera* sp. | | | | | | | | | | | | 21 |
| 58. | *Pentadinium* sp. A | | | | | | | | | | | | |
| 59. | *Selenopemphix brevispinosa*? | | | | | | | | | | | | |
| 60. | *Hystrichosphaeropsis* sp. | | | | | | | | | | | | |
| 61. | *Selenopemphix nephroides* | | | | | | | | | | | | |
| 62. | *Systematophora ?ancyrea* | | | | | | | | | | | | |
| 63. | *Operculodinium*? sp. | | | | | | | | | | | | |
| 64. | *Surculosphaeridium longifurcatum* * | | | | | | | | | | | | |
| 65. | *Phthanoperidinium comatum* * | | | | | | | | | | | | |
| 66. | *Isabelidinium* sp.* | | | | | | | | | | | | |
| | Total Counts | 305 | 298 | 321 | 266 | 313 | 60 | 289 | 54 | 319 | 185 | 314 | 276 |

**Table 4.** Distribution of aquatic palynomorphs in the lower Sarmatian of Babczyn 2 borehole (part 2). (*) indicates reworked taxa; their occurrence is not included in the total count.

| Age | Lower Sarmatian | | | | | | | | | | | | |
|---|---|---|---|---|---|---|---|---|---|---|---|---|---|
| | Machów Formation | | | | | | | | | | | | |
| Lithostratigraphy | Syndesma Beds | | | | | | Krakowiec Clays | | | | | | |
| Sample | 29 | 31 | 33 | 35 | 37 | 39 | 41 | 43 | 47 | 49 | 51 | 53 | 55 |
| *Dinoflagellate Cysts* | | | | | | | | | | | | | |
| 43. *Cordosphaeridium* sp. * | | | | | | | | | | | | | |
| 44. *Batiacasphaera micropapillata* | | | | | | | | | | | | | |
| 45. *Polysphaeridium zoharyi* | | | | | 3 | | | | | | | | |
| 56. *Batiacasphaera hirsuta* | | | | | 1 | | | | | | | | |
| 47. *Membranophoridium aspinatum* * | | | | | | | | | | | | | |
| 48. *Impagidinium pallidum* | | | | | | | | | | | | | |
| 49. *Areosphaeridium michoudii* * | | | | | | | | | | | | | |
| 50. *Operculodinium*? sp. | | | | | | | | | | | | | |
| 51. *Palaeohystrichophora*? sp. * | | | | | | | | | | | | | |
| 52. *Cleistosphaeridium* sp. A * | | | | | | | | | | | | | |
| 53. *Operculodinium* sp. D | | | | | | | | | | | | | |
| 54. *Operculodinium* sp. E | | | | | | | | | | | | | |
| 55. *Operculodinium* sp. F | | | | | | | | | | | | | |
| 56. *Hystrichokolpoma* sp. | | | | | | | | | | 1 | 2 | | |
| 57. *Achomosphaera* sp. | | | | | | | | | | | | | |
| 58. *Pentadinium* sp. A | | | | 8 | | | | | | 1 | | | |
| 59. *Selenopemphix brevispinosa*? | | | | | 2 | 2 | | | | 3 | | 1 | |
| 60. *Hystrichosphaeropsis* sp. | | | | | | 1 | | | | 1 | | | |
| 61. *Selenopemphix nephroides* | | | | | | | 42 | 3 | 3 | 1 | | 4 | 2 |
| 62. *Systematophora ?ancyrea* | | | | | | | 65 | 36 | | | | | |
| 63. *Operculodinium*? sp. | | | | | | | 18 | 21 | 28 | | | 4 | 10 |
| 64. *Surculosphaeridium longifurcatum* * | | | | | | | * | | | | | | |
| 65. *Phthanoperidinium comatum* * | | | | | | | | | | | | | * |
| 66. *Isabelidinium* sp. * | | | | | | | | | | | | | * |
| Total Counts | 0 | 0 | 36 | 313 | 236 | 35 | 159 | 84 | 155 | 274 | 161 | 131 | 103 |

The highest diversity values, between 4.5 and 6.5, were recorded in the lower studied interval, which included the upper Badenian (samples 8–27, except for samples 17 and 25). The diversity decreased in the succession, above the two barren samples (29 and 31), where it oscillated between 2.3 and 4.5 (with a minimal diversity of 0.5 being recorded in sample 51). However, this decreasing upwards diversity trend was not gradual; a characteristic feature was that the diversity in the assemblages from the subsequent samples was alternately higher and lower, thereby forming the zigzag course of the chart (Figure 9). The diversity in dinoflagellate cyst assemblages was generally correlated with the number of taxa. The highest numbers of dinoflagellate cysts (15–22) were recorded in the lower part of the studied interval (samples 9–27). The numbers of taxa from the higher interval were lower, oscillating between 4 and 7, with higher values (i.e., 11, 13, and 16) being present only in three samples (37, 35, and 49, respectively, Figure 9).

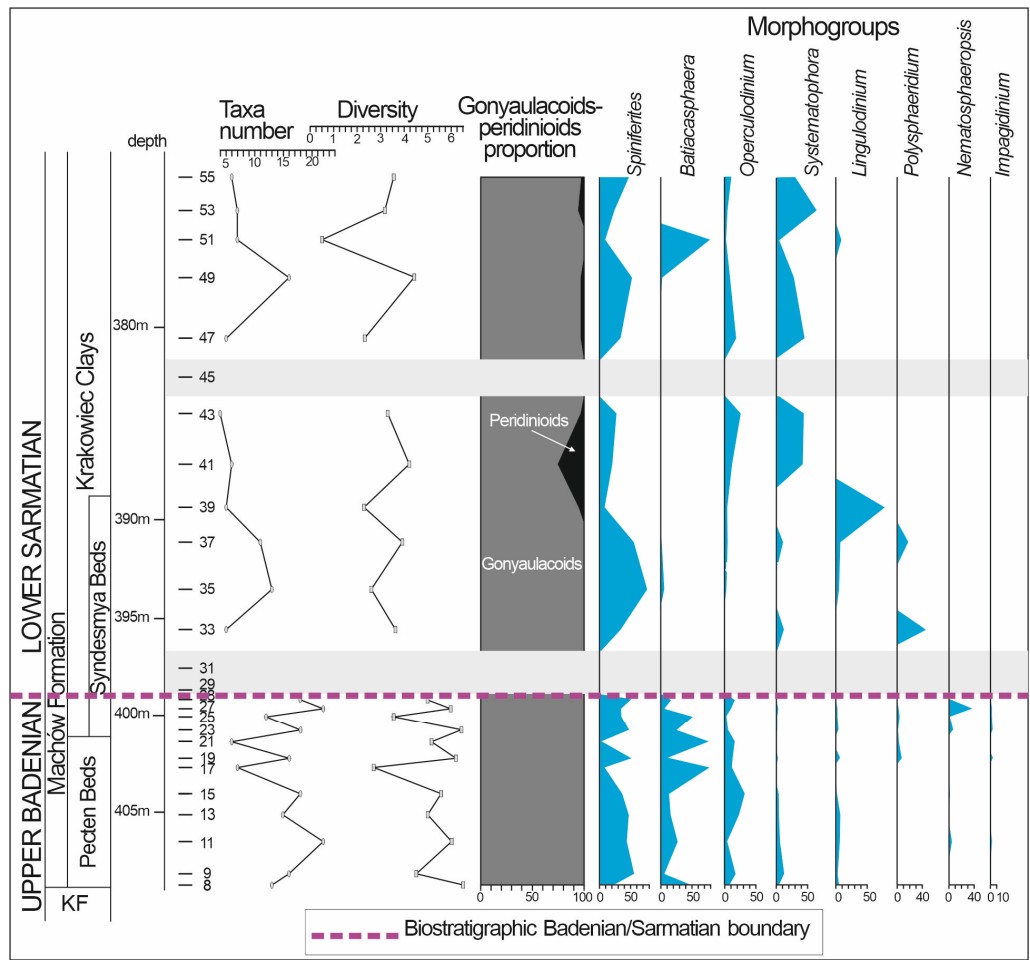

**Figure 9.** Palaeoenvironmental proxies of aquatic palynomorphs from the Babczyn 2 borehole; KF—Krzyżanowice Formation.

Another characteristic feature of the dinoflagellate cyst assemblages from the studied borehole interval was the predominance of gonyaulacoids. Most of the studied samples yielded purely gonyaulacoid assemblages. Peridinioids were represented by *Selenopemphix* and *Lejeunecysta*; the latter and *S. brevispinosa* occurred mainly as single specimens only in a few samples (Tables 1, 2 and 4), whereas *S. nephroides* formed an acme in sample 41 (up to 25% of the whole assemblage, Figure 9).

The diversity and frequency of the taxa in the studied materials were related to the frequencies of some taxa grouped into eight morphogroups. Representatives of the *Spiniferites* and *Batiacasphaera* morphogroups were the most common taxa in the lower part of the studied succession (samples 8–28). Their frequencies showed a negative correlation: samples with common *Spiniferites* yielded few *Batiacasphaera*, whereas increased frequencies of *Batiacasphaera* were associated with decreased *Spiniferites* (Figure 9). This relationship was further correlated with changes in diversity in this interval; *Batiacasphaera*-dominated assemblages (samples 17, 21, and 25) showed the lowest diversities. Representatives of the *Batiacasphaera* morphogroup were virtually absent in the upper interval (Table 1); the only exception was their common occurrence in sample 51, which yielded the most taxonomically impoverished assemblage (Figure 9).

Representatives of the *Operculodinium* morphogroup occurred in all the positive samples except for sample 33, taken just above the two barren samples; their proportions were not high, i.e., rarely exceeding 20% (Figure 9). They showed a negative correlation with the *Batiacasphaera* morphogroup, similar to representatives of the *Spiniferites* morphogroup (Figure 9).

Representatives of the *Systematophora* morphogroup were rare or absent in the lower and middle parts of the studied borehole interval while becoming more frequent in the higher part, starting from sample 41. Their proportions in this interval oscillated between 30 and a bit over 60% (except for sample 51). The proportions of *Systematophora* were negatively correlated with those of *Batiacasphaera* (most visible in sample 51) and *Lingulodinium* (samples 39 and 51) and showed a positive correlation with *Polysphaeridium* (Figure 9). The representatives of the latter were very rare or absent except for two samples (33 and 37), which yielded approximately 46 and 18%, respectively. The *Lingulodinium* share was similar; it was infrequent or absent, being a dominant taxon (almost 80%) in sample 39.

*Nematosphaeropsis labyrinthus* was a very rare species throughout the studied borehole interval; it occurred infrequently in only four samples from the basal part (Table 1). However, it formed an acme in sample 27, collected below the barren interval (samples 29 and 31), where it reached almost 40% of all the dinoflagellate cysts.

The *Impagidinium* morphogroup occurred in the basal part of the studied borehole interval (only one specimen was found in the remaining part, sample 49). The distribution of very uncommon specimens of *Impagidinium* showed a pattern similar to that of *Nematosphaeropsis* (Figure 9).

*4.4. Other Aquatic Palynomorphs*

Dinoflagellate cysts were not the only aquatic palynomorphs found in the studied borehole interval. Other aquatic palynomorphs occurred in all the samples while reaching even higher frequencies in some of them. These included acritarchs (e.g., *Veryhachium*, *Nannobarbophora*, and unidentified small, sphaeromorphic, spiny, and smooth forms); prasinophytes (e.g., Tasmanitaceae and Leiosphaeridiaceae); and zooclasts (foraminifera organic linings, scolecodonts, and remains of presumably arthropods). The diversity of other aquatic palynomorphs exhibited high variability from sample to sample, thereby resembling the distribution pattern of dinoflagellate cyst assemblages.

The lowermost sample (No. 8), yielded frequent Leiosphaeridiaceae (described below as the genus *Leiosphaeridia*, although its precise taxonomic position is tentative). Samples 9–13 yielded foraminifera organic linings and *Nannobarbophora*. *Tasmanites* co-occurred with *Nannobarbophora* in sample 15 and *Leiosphaeridia* in sample 17. Neither sample yielded foraminifera organic linings; these were found in sample 19 (with frequent *Nannobarbophora* and rare *Leiosphaeridia*). The latter had an increased frequency in the subsequent sample, sample 21, which yielded no *Nannobarbophora* and only single foraminifera organic linings. The frequency of Leiosphaearidiaceae in higher samples showed a fluctuating course: it decreased in sample 23, increased in sample 25, and decreased again in sample 27, whereas the occurrences of *Tasmanites* and *Nannobarbophora* in these samples showed the opposite trend. Sample 27 also yielded *Veryhachium*. Rare *Tasmanites* and *Veryhachium* were the only prasinophytes and acritarchs, respectively, in sample 28.

The distribution patterns of acritarchs and prasinophytes in the lower part of the studied borehole interval (samples 8–28) showed specific patterns that were further correlated with dinoflagellate cyst assemblages. There was a clear negative correlation between the assemblages dominated by foraminifera organic linings and *Nannobarbophora* (and *Tasmanites* to a lesser degree) and those dominated by *Leiosphaeridia*. Moreover, the samples that yielded the former assemblages yielded relatively diverse dinoflagellate cyst assemblages. These samples, which yielded impoverished assemblages dominated by *Batiacasphaera*, also exhibited increased frequencies of *Leiosphaeridia*.

Various spherical forms attributed to Leiosphaeridiaceae became the nearly sole aquatic palynomorphs in samples 29 and 31, which yielded no dinoflagellate cysts; scolecodonts were found in sample 31. *Leiosphaeridia* were again dominant in the subsequent sample (33), which yielded impoverished dinoflagellate cyst assemblages with relatively frequent *Polysphaeridium*.

Higher intervals of the studied borehole (samples 35–55) yielded common to very common Leiosphaeridiaceae, represented by morphologically diversified forms, some of

which likely represent sphaeromorphic, smooth acritarchs. They were predominant in these samples over other prasinophytes, and acritarchs were commonly the only aquatic palynomorphs (other than dinoflagellate cysts). Rare *Nannobarbophora* occurred in samples 35, 37, and 49, and a single foraminifera organic lining was found in sample 51.

Reworked dinoflagellate cysts commonly occurred in the lower part of the studied borehole interval (samples 8–28). These were Palaeogene specimens representing mainly Eocene–Oligocene forms, and the *Palaeohystrichophora* found in sample 27 may have been Cretaceous forms. Higher borehole intervals yielded much less frequently reworked specimens, which were mainly Cretaceous forms (Tables 1–4; reworked taxa are highlighted with an asterisk).

### 4.5. Spore–Pollen Analysis

All the samples yielded pollen grains and spores suitable for palynological analysis. In most samples, 300–500 pollen grains and spores (a minimum of 100 in sample 21) were identified. Besides sporomorphs, all the co-occurring non-pollen palynomorphs in the samples were counted. Most sporomorphs were poorly preserved; however, they were still identifiable. A total of 71 fossil-species (including nine species of plant spores, 16 species of gymnosperm pollen, and 46 species of angiosperm pollen) were identified (Table 5). In all the samples, among the pollen grains of the gymnosperms, bisaccate pollen grains (Figure 6C) of the Pinaceae family (mainly *Pinus*, plus *Cathaya*, *Abies*, *Picea*, *Keteleeria/Pseudolarix*, and *Cedrus*); Taxodium/Glyptostrobus (Figure 6D); *Tsuga* (Figure 6E,F); and *Sciadopitys* (Figure 6G,H) were the most common. Additionally, several pollen grains of *Sequoia/Sequoiadendron/Metasequoia/Cryptomeria* were encountered (Figure 10).

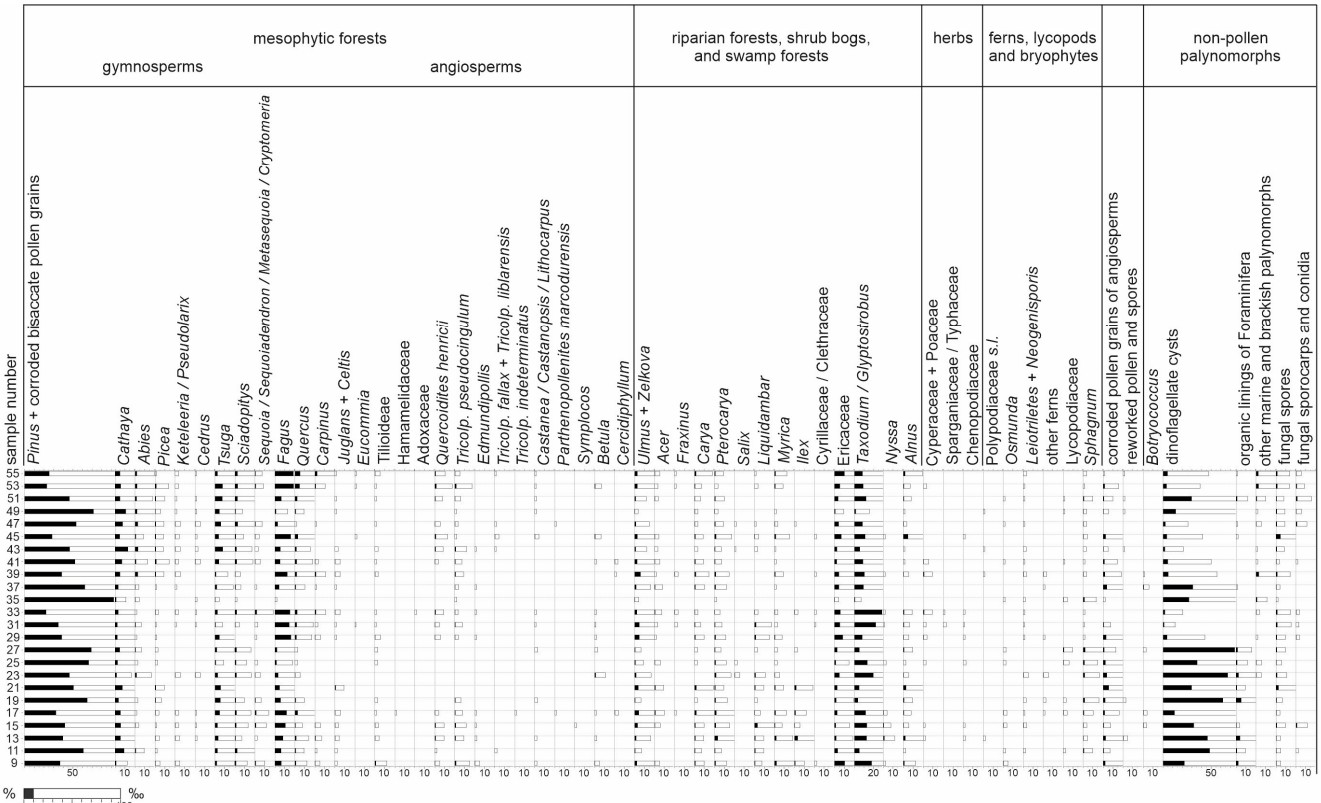

**Figure 10.** Palynological diagram for the Babczyn 2 borehole. Black bars show percentages (%); white bars show percentages ×10 (‰).

**Table 5.** Spores and pollen grains recorded in samples from the Babczyn 2 borehole. Taxonomies and botanical affinities according to [53–56]. The following palaeofloristical elements have been distinguished: "palaeotropical" (P), including "tropical" (P1) and "subtropical" (P2), and "arctotertiary" (A), including "warm-temperate" (A1) and "temperate" (A2), as well as cosmopolitan (P/A).

| Fossil taxon | Botanical affinity | Element |
|---|---|---|
| *Spores of plants* | | |
| *Baculatisporites* sp. | Osmundaceae: *Osmunda* | P/A |
| *Camarozonosporites* sp. | Lycopodiaceae: *Lycopodiella* sect. *Campylostachys* | P |
| *Distverrusporis* sp. | Sphagnaceae: *Sphagnum* | P/A |
| *Laevigatosporites* sp. | Polypodiaceae, Davalliaceae, and other ferns | P/A |
| *Leiotriletes* sp. | Lygodiaceae and other ferns | P |
| *Neogenisporis* sp. | Gleicheniaceae and Cyatheaceae | P1 |
| *Retitriletes* sp. | Lycopodiaceae: *Lycopodium* | A |
| *Stereisporites* sp. | Sphagnaceae: *Sphagnum* | P/A |
| *Verrucatosporites* sp. | Davalliaceae, Polypodiaceae, and other ferns | P/A |
| *Pollen grains of gymnosperms* | | |
| *Abiespollenites* sp. | Pinaceae: *Abies* | A |
| *Cathayapollis* sp. | Pinaceae: *Cathaya* | A1 |
| *Cedripites* sp. | Pinaceae: *Cedrus* | A1 |
| *Inaperturopollenites concedipites* (Wodehouse and Krutzsch) | Cupressaceae: *Taxodium* and *Glyptostrobus* | P2/A1 |
| *Inaperturopollenites verrupapillatus* Trevisan | Cupressaceae: *Taxodium* and *Glyptostrobus* | P2/A1 |
| *Keteleeriapollenites* sp. | Pinaceae: *Keteleeria* and *Pseudolarix* | A1 |
| *Piceapollis praemarianus* Krutzsch | Pinaceae: *Picea* | A |
| *Piceapollis* sp. | Pinaceae: *Picea* | A |
| *Pinuspollenites labdacus* (Potonié) Raatz | Pinaceae: *Pinus sylvestris* type | A |
| *Pinuspollenites* sp. | Pinaceae: *Pinus* | A |
| *Sciadopityspollenites verticillatiformis* (Zauer) Krutzsch | Sciadopityaceae: *Sciadopitys* | A1 |
| *Sciadopityspollenites* sp. | Sciadopityaceae: *Sciadopitys* | A1 |
| *Sequoiapollenites* sp. | Cupressaceae: *Sequoia*, *Sequoiadendron*, and *Metasequoia* | A1 |
| *Zonalapollenites gracilis* Krutzsch | Pinaceae: *Tsuga* | A |
| *Zonalapollenites verrucatus* Krutzsch | Pinaceae: *Tsuga* | A |
| *Zonalapollenites* sp. | Pinaceae: *Tsuga* | A |
| *Pollen grains of angiosperms* | | |
| *Aceripollenites* sp. | Sapindaceae: *Acer* | A1 |
| *Alnipollenites verus* Potonié | Betulaceae: *Alnus* | P2/A |
| *Caprifoliipites* sp. | Adoxaceae: *Sambucus* and *Viburnum* | P/A1 |
| *Carpinipites carpinoides* (Pflug) Nagy | Betulaceae: *Carpinus* | P2/A1 |
| *Caryapollenites simplex* (Potonié) Raatz | Juglandaceae: *Carya* | A1 |
| *Celtipollenites* sp. | Ulmaceae: *Celtis* | P/A1 |
| *Cercidiphyllites minimireticulatus* (Trevisan) Ziembińska-Tworzydło | Cercidiphyllaceae: *Cercidiphyllum* | A1 |
| *Chenopodipollis* sp. | Amaranthaceae (incl. Chenopodiaceae) | P/A |

**Table 5.** *Cont.*

| | | |
|---|---|---|
| *Cupuliferoipollenites oviformis* (Potonié) Potonié | Fagaceae: *Castanea*, *Castanopsis*, and *Lithocarpus* | P2/A1 |
| *Cupuliferoipollenites pusillus* (Potonié) Potonié | Fagaceae: *Castanea*, *Castanopsis*, and *Lithocarpus* | P2/A1 |
| *Cyperaceaepollis neogenicus* Krutzsch | Cyperaceae | P/A |
| *Cyrillaceaepollenites megaexactus* (Potonié) Potonié | Cyrillaceae and Clethraceae | P |
| *Edmundipollis* sp. | Araliaceae, Cornaceae, and Mastixiaceae | P/A |
| *Ericipites baculatus* Nagy | Ericaceae | A |
| *Ericipites ericius* (Potonié) Potonié | Ericaceae | A |
| *Ericipites* sp. | Ericaceae | A |
| *Eucommiapollis minor* Menke | Eucommiaceae: *Eucommia* | A1 |
| *Faguspollenites* cf. *verus* Raatz | Fagaceae: *Fagus* | A |
| *Faguspollenites* sp. | Fagaceae: *Fagus* | A |
| *Fraxinipollis oblatus* Słodkowska | Oleaceae: *Fraxinus* | P/A |
| *Graminidites* sp. | Poaceae: Pooideae | P/A |
| *Ilexpollenites iliacus* (Potonié) Thiergart | Aquifoliaceae: *Ilex* | P/A1 |
| *Ilexpollenites margaritatus* (Potonié) Thiergart | Aquifoliaceae: *Ilex* | P2 |
| *Intratriporopollenites* sp. | Malvaceae: Tilioideae | P/A1 |
| *Juglanspollenites* sp. | Juglandaceae: *Juglans* | A1/P2 |
| *Myricipites* sp. | Myricaceae: *Myrica* | P2/A1 |
| *Nyssapollenites* sp. | Nyssaceae: *Nyssa* | P2/A1 |
| *Parthenopollenites marcodurensis* (Pflug and Thomson) Traverse | Vitaceae | P/A1 |
| *Periporopollenites orientaliformis* (Nagy) Kohlman-Adamska and Ziembińska-Tworzydło | Altingiaceae: *Liquidambar* | A1 |
| *Periporopollenites stigmosus* (Potonié) Thomson and Pflug) | Altingiaceae: *Liquidambar* | A1 |
| *Polyatriopollenites stellatus* (Potonié) Pflug | Juglandaceae: *Pterocarya* | A1 |
| *Quercoidites henricii* (Potonié) Potonié, Thomson, and Thiergart) | Fagaceae: *Quercus* | P2/A1 |
| *Quercopollenites rubroides* Kohlman-Adamska and Ziembińska-Tworzydło | Fagaceae: *Quercus* | A1 |
| *Quercopollenites* sp. | Fagaceae: *Quercus* | A1 |
| *Salixipollenites capreaformis* Planderová | Salicaceae: *Salix* | A |
| *Salixipollenites* sp. | Salicaceae: *Salix* | A |
| *Sparganiaceaepollenites* sp. | Sparganiaceae and Typhaceae | P/A |
| *Symplocoipollenites vestibulum* (Potonié) Potonié | Symplocaceae: *Symplocos* | P |
| *Tricolporopollenites fallax* (Potonié) Krutzsch | Fabaceae | P/A |
| *Tricolporopollenites indeterminatus* (Romanowicz) Ziembińska-Tworzydło | Hamamelidaceae | P2 |
| *Tricolporopollenites liblarensis* (Thomson) Hochuli | Fabaceae | P/A |
| *Tricolporopollenites pseudocingulum* (Potonié) Thomson and Pflug | Fagaceae? and Styracaceae? | P/A1 |

**Table 5.** *Cont.*

| *Trivestibulopollenites betuloides* Pflug | Betulaceae: *Betula* | A |
| *Ulmipollenites undulosus* Wolff | Ulmaceae: *Ulmus* | A2 |
| *Ulmipollenites* sp. | Ulmaceae: *Ulmus* | A2 |
| *Zelkovaepollenites* sp. | Ulmaceae: *Zelkova* | A1 |

Among the angiosperm pollen grains, *Fagus* (Figure 6I), Ericaceae (Figure 6L, M), *Quercus* (fossil-genus *Quercopollenites*—Figure 6K—and fossil-species *Quercoidites henricii*), *Ulmus* (Figure 6J), *Pterocarya*, *Carya*, *Zelkova*, *Liquidambar* (Figure 6N), *Alnus*, and *Myrica* were the most common. Pollen grains of *Acer*, *Carpinus*, fossil-species *Tricolporopollenites pseudocingulum*, Tilioideae, *Betula*, *Celtis*, and *Juglans* were regularly observed in the samples. *Ilex* and *Nyssa* were mainly present in samples from the lower section of the profile. In addition, a few pollen grains of Adoxaceae, *Castanea/Castanopsis/Lithocarpus*, *Cercidiphyllum*, Cyrillaceae/Clethraceae, *Eucommia*, *Fraxinus*, *Salix*, and *Symplocos*, as well as the fossil-species *Edmundipollis* sp. (Figure 6O), *Parthenopollenites marcodurensis*, *Tricolporopollenites fallax*, *T. indeterminatus*, and *T. liblarensis* were encountered. Herbs were very rare, represented by members of the Chenopodiaceae, Cyperaceae, Poaceae, and Sparganiaceae/Typhaceae families (Figure 10, Table 5).

Fern spores were also rare, with the *Baculatisporites*, *Laevigatosporites*, *Leiotriletes*, *Neogenisporis* (Figure 6B), and *Verrucatosporites* fossil genera being present. In contrast, spores of *Sphagnum* (Figure 6A) were encountered in all the samples. Several lycopod spores (related to the modern genera *Lycopodium* and *Lycopodiella*) were also recorded. Typical freshwater algae were virtually absent, and only scarce *Botryococcus* colonies were encountered. Microremains of fungi (Figure 6P,Q) were continually found in the palynological profile. Fungal spores were present in all the samples, whereas sporocarps were mainly present in the upper section of the profile (Figure 10).

In all the samples, "arctotertiary" and cosmopolitan taxa prevailed, but "palaeotropical" and "palaeotropical/warm temperate" taxa were also present (Table 5). "Palaeotropical" elements were represented by single specimens of *Camarozonosporites* sp., *Leiotriletes* sp., *Neogenisporis* sp., *Cyrillaceaepollenites megaexactus*, *Ilexpollenites margaritatus*, *Symplocoipollenites vestibulum*, and *Tricolporopollenites indeterminatus*. The representation of the "palaeotropical/warm temperate" taxa was more significant, and pollen grains of *Inaperturopollenites concedipites*, *I. verrupapillatus*, *Cupuliferoipollenites oviformis*, *C. pusillus*, *Ilexpollenites iliacus*, *Myricipites* sp., *Nyssapollenites* sp., *Parthenopollenites marcodurensis*, *Quercoidites henricii*, and *Tricolporopollenites pseudocingulum* occurred.

## 5. Discussion

### 5.1. Aquatic Palaeoenvironment

The qualitative and quantitative analyses of the foraminifera and aquatic palynomorphs showed that the studied borehole interval accumulated under variable, unstable sedimentary conditions (Figure 11).

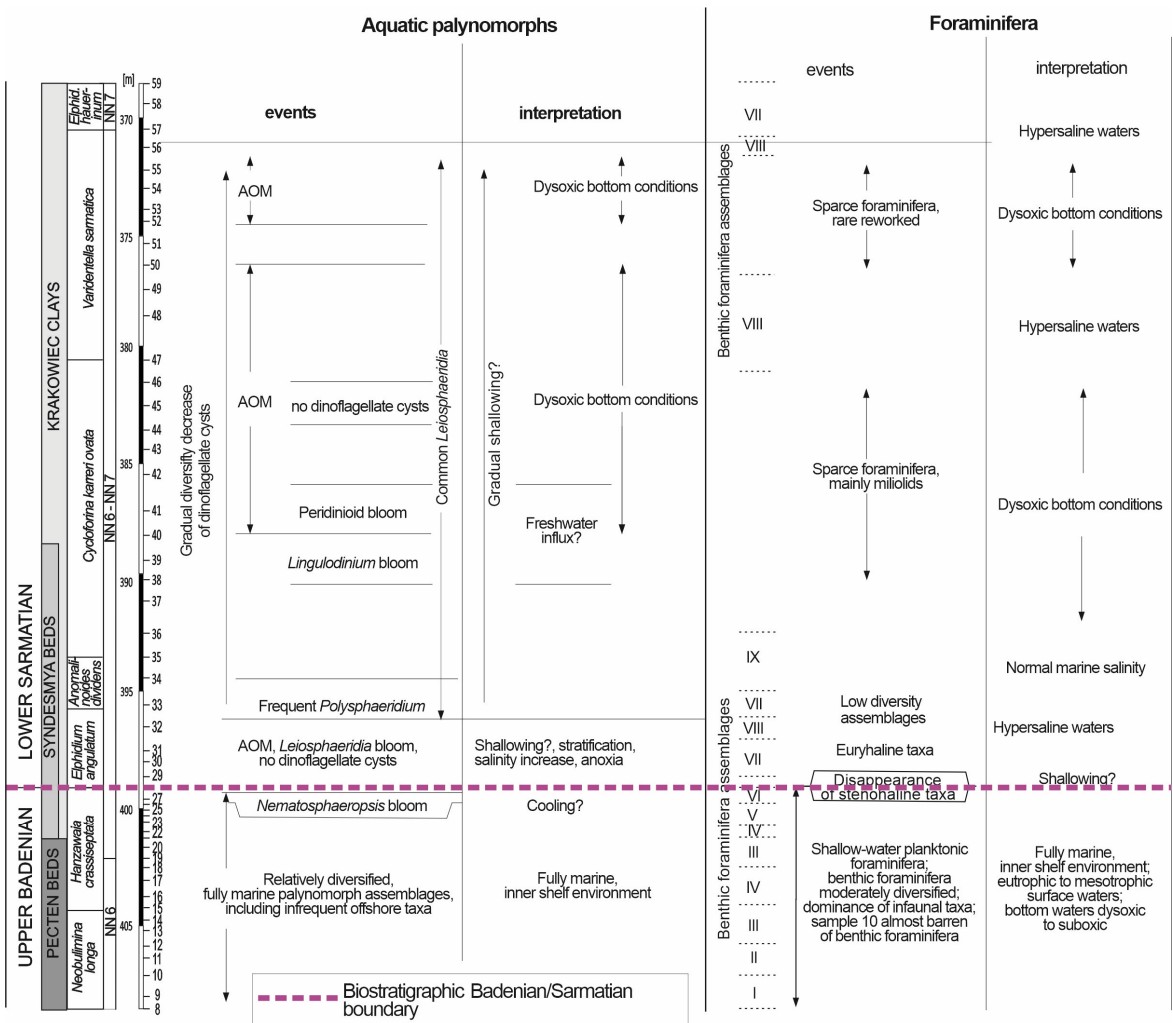

**Figure 11.** Palaeoenvironmental changes in the late Badenian and early Sarmatian based on aquatic palynomorph and foraminiferal data from the Babczyn 2 borehole.

### 5.1.1. Foraminiferal Record

Foraminifera are commonly used as proxies for palaeoenvironmental studies because of the correlation between foraminiferal test shapes and their palaeoenvironmental requirements [32,58]. Planktonic foraminifera can be useful indicators of ancient sea-level changes because of their depth stratification [44,59,60]. The vertical distribution of planktonic foraminifera is directly related to their life cycle. They reproduce at species-specific depths relative to the pycnocline and under distinct temperature and salinity conditions; thus, they require a water column at certain depths for ontogenetic vertical migration and reproduction [60,61]. Simple morphologies (r-strategists), which are representative of the most cosmopolitan and opportunistic taxa, inhabit shallow, more nutrient-rich, eutrophic waters [62,63]. Planktonic foraminifera in the upper Badenian part of the succession included *Globigerina* (with *Globigerina bulloides* being dominant within this group) and rarely recorded *Trilobatus* and *Velapertina*. *Globorotalia*, which shows preferences for deeper/intermediate habitats, was not recorded in the studied succession. *Globigerina bulloides* reproduces primarily within the upper 60 m of the water column and exhibits the maximum abundance at this depth [60,64–66]. It is also considered to be an indicator of cooler [67,68] and nutrient-rich eutrophic waters [44,62,69,70]. *Velapertina* was endemic to the Central Paratethys, where it appeared to have repeated the same sequence of transitional steps as *Orbulina* [48]. The two genera had broadly overlapping distributions and occupied a niche similar to that of *Orbulina* in the surface waters [48].

The ratio between the planktonic and benthic foraminifera (P/B) is related to the water depth, and the percentage of planktonic foraminifera generally increases with increasing distance from the shore. However, next to the water depth, the oxygen level of the bottom water has profound effects on the abundance and diversity of benthic foraminifera. This significantly influences the percentage of planktonic foraminifera in assemblages [50]. According to the foraminiferal trophic condition and oxygen concentration (TROX) model [71], the benthic foraminifera distribution is a function of the interplay between the food availability and oxygen concentrations. The dominance of infaunal species is interpreted as an indicator of an increase in the organic matter supply and the dominance of the eutrophic and dysoxic environments. In oligotrophic and well-oxygenated environments, assemblages are dominated by epifaunal species [29,72]. The variations in the proportions of epifaunal and infaunal species indicate distinctive inputs of organic matter (phytodetritus input *versus* bacterial activity, respectively) [73].

Benthic foraminiferal assemblages from the Badenian part of the studied section (Assemblages from I to VI) are dominated by deep infaunal buliminids (*Bulimina*) and shallow infaunal uvigerinids (*Uvigerina* and *Angulogerina*). *Melonis* and *Sphaeroidina*—shallow infaunal species—as well as epifaunal *Cibicidoides*, *Hanzawaia*, and *Heterolepa* are common components of benthic foraminiferal assemblages. The high dominance of infaunal taxa (>95%), mainly buliminids and uvigerinids (Assemblage I) in the lowermost part of the studied interval (samples from 8 to 10), indicates a large supply of organic matter to the sea floor, the dominance of eutrophic conditions, and the oxygen impoverishment of the bottom waters in this area. In the basal part of the studied interval, benthic foraminifera were very rare in sample 10 (Figure 7). Their absence can be explained by a short episode of dysoxic conditions at the bottom of the sea at this site.

The gradual increase in epifauna up the section from 20% in Assemblage II to almost 40% in Assemblage VI indicates a change from eutrophic to mesotrophic conditions, a decrease in the organic matter supply, and an amelioration in oxygen in the bottom waters.

The alternation in the assemblages dominated by infaunal and shallow infaunal taxa, *Bulimina*, *Angulogerina*, and, to lesser degrees, *Bolivina* (Assemblages III, IV, and VI) and *Uvigerina* (Assemblages II and V), may be the result of slightly different requirements in the types of food and oxygen levels of each taxon. In the uppermost Badenian (samples from 11 to 28), the contribution of the infaunal taxa decreases, and diversity increases, which suggest mesotrophic conditions in the surface waters and increased oxygenation in the bottom waters.

There is also one more factor that could influence the composition of benthic foraminiferal assemblages. Middle Miocene tectonic activity in the Carpathian Foredeep [74] resulted in intensive volcanism with enhanced input of volcanoclastic material [3,75]. Studies and monitoring of the changes in the compositions of benthic foraminiferal assemblages from the South China Sea following the 1991 Mt. Pinatubo eruption have revealed that high tephra settling rates through the water column led to the mass mortality of benthic biota and to significant survival and rapid recovery of the ecosystem in distal parts of the ash fan, where the thickness of the ash layer did not exceed a few millimetres [76,77].

The Badenian/Sarmatian boundary, as correlated with a sudden extinction of stenohaline foraminifera (BSEE), is placed just above sample 28 (within the lowermost part of the Syndesmya Beds; Figure 7). The overlying sample (29) is almost barren of foraminifera from both groups, planktonic and benthic ones. Higher up in the section, assemblage VII is dominated by elphidiids alternating with assemblage VIII, which is dominated by miliolids. These groups of foraminifera are euryhaline and can tolerate a wide range of salinities from brackish to elevated. Keeled species of *Elphidium* prefer inner-shelf environments characterised by salinities of 30–70‰ and 0–50 m water depths. At this site, they are represented only by very rare specimens of *Elphidium fichtelianum* and *E. macellum*. Unkeeled species occur in brackish–hypersaline marshes and lagoons showing salinity ranges of 0–70‰ [29,38]. In the studied section, they are represented by *Elphidium excavatum*, *E. angulatum*, *E. advenum*, and *E. hauerinum*. Miliolids (*Quinqueloculina* and *Triloculina*)

occur in hypersaline marine environments (salinity ranging from 32 to 55–65‰), mainly in hypersaline lagoons or on the marine inner shelf [29,78]. The replacement of stenohaline foraminiferal assemblages by euryhaline ones indicates a salinity increase, while the disappearance of planktonic foraminifera suggests a shallowing of the sea.

The next almost monospecific assemblage (assemblage IX), which is dominated by *Anomalinoides dividens*, probably reflects a salinity lowering to normal marine values. Presently, marine environments are characteristic for *Anomalinoides* [79], although in the past, its wider environmental range, including brackish salinity conditions, was accepted [46] (with references therein).

### 5.1.2. Palynomorph Record

The presence of marine palynomorphs suggests that most of the studied section accumulated in marine environments. However, changes in palynofacies and the compositions of aquatic palynomorph assemblages among particular samples suggest that these environmental conditions were unstable while being subjected to numerous changes related to sea-level fluctuations, salinity, and climatic changes. All these factors may have played important roles in the aforementioned changes in aquatic assemblages.

Undoubtedly, the lower part of the studied borehole interval (samples 8–28; the Pecten Beds and the lowermost part of the Syndesmya Beds) accumulated in a marine environment influenced by runoff from the neighbouring land. The upward-decreasing proportions of cuticles suggest that land influences were gradually decreasing; the simultaneous increase in pollen grains (Figure 8) confirms this interpretation, as most of the grains were bisaccate forms, which are airborne, and can be transported for long distances (the so-called 'Neves effect'; see, e.g., [80,81]). However, there were no other signs of land influences, such as a salinity decrease; there were neither freshwater algae nor dinoflagellate cysts, which prefer low-salinity waters. Despite the relatively intense land influence, this interval was presumably deposited under relatively offshore conditions compared with the interval above sample 28. Only this interval yielded *Impagidinium* and *Nematosphaeropsis*, two dinoflagellate cyst genera commonly associated with offshore waters (e.g., [82–84]), while the upper interval yielded only a single *Impagidinium* (sample 49; Table 1).

The interval below the Badenian/Sarmatian boundary (samples 17–28) showed a series of environmental fluctuations manifested by alternating changes in aquatic palynomorph compositions. High-diversity marine palynomorph assemblages dominated by *Spiniferites* alternated with less-diverse *Batiacasphaera*-dominated assemblages. Moreover, *Nematosphaeropsis labyrinthus* occurred during this interval and exhibited a negative correlation with *Batiacasphaera* (Figure 9).

*Spiniferites* (mainly *S. ramosus* and the morphologically similar *Achomosphaera*) occurs in a broad marine environmental spectrum and is commonly treated as a cosmopolitan taxon (e.g., [82]). But several authors, such as in [85,86], have suggested that its frequency increases offshore. It is also present in the marine Middle Miocene strata of the Carpathian Foredeep and is the most frequent in marine shelf environments (e.g., [87]). This suggests that the samples that yielded diversified assemblages (with frequent *Spiniferites*) represent strata accumulated in a marine environment, representing an offshore setting, in contrast to those that yielded impoverished assemblages with common *Batiacasphaera*. Although the latter genus was described by [88] as a *Batiacasphaera micropapillata* complex, including *B. micropapillata* and *B. minuta*, as is typical for outer neritic to oceanic waters with slightly increased salinities, *Batiacasphaera* (mainly *B. sphaerica*) tends to exhibit increased frequencies in restricted environments possibly associated with water shallowness and/or increased salinity [87,89]. The negative correlation between *Batiacasphaera* and *Nematosphaeropsis labyrinthus* may be an additional clue suggesting sea-level fluctuations during this period. However, *Nematosphaeropsis labyrinthus* is treated as a cold-water species (e.g., [90]) in contrast to *Batiacasphaera*, which is believed to be a temperate- to warm-water taxon [88]. This suggests that an acme of *N. labyrinthus* just below the Badenian/Sarmatian boundary may be the result of a temperature drop in sea-surface waters (a similar inverse correlation

between *N. labyrinthus* and *Batiacasphaera micropapillata* was noted by [91] and was interpreted as a possible cooling). Notably, high frequencies of *N. labyrinthus* were found in the Pecten Beds overlying the Badenian evaporites in the Carpathian Foredeep [92]. This might indicate a cooling period (or periods) after the accumulation of the evaporitic series. However, *N. labyrinthus* is known to be found in warm-water middle Badenian Korytnica Clays and their offshore age equivalents—the Skawina Beds [87,89].

The interval that occurred just above (samples 29 and 31) was characterised by a lack of dinoflagellate cysts and the flowering of Leiosphaeridiaceae. Any of the aforementioned environmental factors—albeit at a larger magnitude—could be the one that led to such a drastic environmental change in the earliest Sarmatian, associated with the collapse of the dinoflagellate floras. The precise reconstruction of environmental changes during the accumulation of these strata is difficult. The absence of dinoflagellate cysts (Figure 8) indicated disastrous conditions for dinoflagellate cysts. However, the most likely reason was that the salinity increased above the tolerable level, even for hypersaline forms (e.g., *Polysphaeridium*) but was still favourable for *Leiosphaeridia*. This Prasinophyta genus (e.g., [93]) was described in the Carpathian Foredeep, more specifically, in the middle Badenian evaporitic strata accumulated under increased water salinity conditions (e.g., [89]) and in upper Badenian stress deposits [94]. These possible hypersaline conditions were associated with stagnant, possibly stratified, waters that led to anoxic conditions in the bottom waters, as evidenced by AOM (Figure 8). Hypersaline waters are usually stagnant. If waters are affected by strong circulation, particularly in a deeper basin, then hypersalinar conditions are less likely to appear. The Palynological record shows blooms of *Leiosphaera*, a prasinophyte alga widespread in the Miocene salt deposits of the Carpathian Foredeep, simultaneously with the appearance of amorphous organic matter (AOM), which is typical for anoxic bottom conditions. We have no indications as to the precise salinity level (certainly above 3.5%).

The cessation of these conditions was caused by a possible sea-level rise and the gradual return of a less-saline water regime. The latter interpretation is supported by the fact that sample 33 yielded a high frequency of *Polysphaeridium* (a genus known to exist in hypersaline environments, e.g., [95]), which benefited from the transitional salinity levels between the highly saline conditions that were disastrous for the genus (samples 29 and 31) and the conditions that prevailed during the accumulation of the higher interval (samples 35 and above). *Polysphaeridium* was described at a similar position, i.e., in the strata directly overlying the chemical deposits in other areas of the Miocene of the Carpathian Foredeep (e.g., [94,96–98]).

The higher interval (samples 35–55, i.e., the upper part of the Syndesmya Beds and the overlying part of the Krakowiec Clays; Figure 9) was deposited under more restricted and variable environmental conditions (compared to those of the basal interval, namely, samples 8–28).

This interval yielded no offshore species (except for a single *Impagidinium* specimen in sample 49), and the diversity and number of taxa decreased significantly (Figure 9). However, the factors responsible for these conditions remain unclear. There were no clear indicators of sea-level or salinity changes. An important event was the disappearance of *Batiacasphaera*, which can be interpreted either as the result of cooling or changes in sea-water chemistry; however, its nature remains uncertain. A drop in the sea-surface water temperature is also suggested by the lack of *Nannobarbophora gedlii*, which is believed to be a warm-water species [99].

Some recorded events may indicate fluctuations in the salinity level. There are almost no stenohaline foraminiferal organic linings (present in the lower interval, namely, samples 8–28), and some taxa known to benefit from low-salinity, commonly nutrient-rich waters, show temporarily enriched frequencies (e.g., [84]). These include *Lingulodinium machaerophorum* and heterotrophic Congruentidiaceae (peridinioids are represented in the present material by *Selenopemphix* and *Lejeunecysta*). *L. machaerophorum* dominated sample 39, whereas rare peridinioids occurred in the top interval (samples 39–55), forming an

acme of *Selenopemphix* in sample 41 (Table 4; Figure 9). This interval, in turn, was preceded by samples 33 and 37, which yielded *Polysphaeridium*—a hypersaline taxon (see above). Another characteristic feature of the strata above the Syndesmya Beds was the common occurrence of *Systematophora*. This genus was missing or rarely exceeded 10% in the lower interval, whereas it accounted for over 40%, maximally over 60%, in some samples from the upper interval (Figure 9). However, palaeoenvironmental preferences of *Systematophora* are poorly understood. Ref. [86] included *Systematophora placacantha* in the *Glaphyrocysta* eco-group, which included open-marine (i.e., fully marine) and warm-water taxa. Ref. [82] highlighted the widespread occurrence of this genus and linked it to shelf environments in any climatic setting. The analysis of *Systematophora* (*S. placacantha* and *S. ancyrea*) occurrences in the Miocene of the Carpathian Foredeep shows a broad environmental tolerance range of this genus, as it is a member of fully marine (e.g., [87,100]) and impoverished assemblages (e.g., [92]). However, a noticeable feature of the occurrence of this genus is the remarkable frequency increase in Sarmatian deposits (e.g., [92]), which may be linked to the euryhaline nature of this genus. A similar acme of *Systematophora* (as *Clesitosphaeridium*) was noted in the uppermost Badenian–lower Sarmatian Paratethyan deposits in Hungary [101].

Another clue suggestive of fluctuating, mainly restricted, environments during the accumulation of the upper part of the Syndesmya Beds and the Krakowiec Clay is the persistent presence of *Leiosphaeridia* (and presumably some other sphaeromorphic, smooth acritarchs). Their occurrence may reflect not only salinity fluctuations (see above) but also generally cooler surface waters and/or increased nutrient availability (see [91], p. 55 and references therein).

An exception was an assemblage yielded from sample 51, i.e., a low-diversity assemblage dominated by *Batiacasphaera*, with no peridinioids. Its palynofacies is also exceptional as it includes no AOM in contrast to the remaining samples from this interval. The appearance of *Batiacasphaera* may be related to the temporal return of fully marine conditions, possibly because of a short-term sea-level rise. The AOM that occurred in most of the samples from this interval indicates stagnant waters with a low oxygen content in the bottom environments. Even a low-magnitude sea-level rise could induce sea-water circulation and better ventilation at the bottom. The appearance of *Batiacasphaera* in sample 51 may also reflect an increase in surface sea-water temperatures.

*5.2. Terrestrial Palaeoenvironment*

Although some sporomorphs are corroded, the palynological analysis provided valuable information about the palaeovegetation, palaeogeography, and palaeoclimate. The degree of pollen grain destruction can also be used as a source of information. For example, some bisaccate pollen grains were heavily corroded, particularly in samples with high marine palynomorph contents (at the lower part of the section; Figure 10). This pollen could have easily been transported over long distances, from both the northern seashores and the Carpathians, and their abundance tends to increase offshore (the so-called 'Neves effect'). Nevertheless, conifers likely played an important role in the coastal forests. For example, the better-preserved non-bisaccate pollen grains of *Tsuga* and *Sciadopitys* were the most likely components of mesophytic forests in the vicinity. In the upper section of the profile (above sample 27), some pollen grains were found in clumps (Ericaceae, *Fagus*, and *Salix*), which may also indicate the proximity of their habitats to the coast.

The results of the palynological analysis revealed the presence of mesophytic and wetland vegetation along the Paratethys shoreline during sedimentation. Mesophytic forests were composed of *Fagus*, *Quercus* (also thermophilous oaks producing pollen of the fossil-species *Quercoidites henricii*), *Tsuga* and other conifers, *Carpinus*, Tilioideae, and others, with a relatively small admixture of Castaneoideae and other thermophilous taxa. *Ulmus*, *Pterocarya*, *Carya*, *Zelkova*, *Liquidambar*, *Alnus*, *Acer*, *Fraxinus*, and *Salix* probably grew in riparian forests in periodically flooded areas, such as river valleys. *Taxodium* and/or *Glyptostrobus*, together with *Alnus* and *Nyssa*, were likely elements of swamp forests growing in permanently flooded areas in the vicinity. Similarly, *Osmunda* and other ferns

grew in wet places. Various shrubs of the Ericaceae family, as well as *Myrica* and *Ilex*, probably were components of shrub communities. The abundance of ericaceous plants may be related to open areas adjacent to the coastline within coastal peat bogs or heathland-type shrub vegetation and partly with forest undergrowth [102,103]. Herbs were represented mainly by the Cyperaceae, Sparganiaceae/Typhaceae, and Poaceae families, and at least some of them could have grown in freshwater margins (e.g., Cyperaceae, *Sparganium*, and/or *Typha*). The presence of numerous pollen grains of Chenopodiaceae coincides with the occurrence of high-salinity habitats, such as seashores, because many of the recent species of this family are halophytes that tolerate salty soils.

The changes in the frequency of the spore–pollen taxa were rather small, and no rapid change in vegetation was observed. Nevertheless, the lower section appeared richer in "palaeotropical" taxa (*Ilexpollenites margaritatus*, *Symplocoipollenites vestibulum*, and *Tricolporopollenites indeterminatus*), although it generally had a high content of marine palynomorphs, and the land elements have had little chance of reaching this location. In contrast, the upper section contained more diverse Pinaceae pollen grains, including *Abies* and *Picea*, which can be interpreted as a manifestation of the mountain forest development. Such changes are gradual and modified by the changing sea influences; therefore, a boundary between the levels cannot be set.

Diverse fungal microremains (i.a., *Asterosporium*, *Cephalothecoidomyces*, *Diporotheca*, *Phragmothyrites*, *Potamomyces*, and cf. *Tetraploa*) were found, mainly in the upper part of the profile. Their presence indicated a brackish environment and their proximity to the seashore. *Cephalothecoidomyces* G. Worobiec, Neumann, and E. Worobiec and *Potamomyces* K.D. Hyde suggest that abundant decaying wood could have accumulated in humid and swampy places [104–106]. Contemporary *Diporotheca webbiae* D. Hawksworth., B. van Geel, and P. Wiltshire, similar to the fossil specimen from Babczyn, is associated with alder carrs [107]. Therefore, it is probable that the coastal areas, at least in some parts, were overgrown by swamp forests, as also indicated by the studied fossil pollen grains. The co-occurrence of the fossil conidia of the *Asterosporium asterospermum* (Pers.) Hughes fungus with *Fagus* pollen grains indicates that beech trees grew close to the seashore [108].

The results of the palynological analysis indicated that the climate during the deposition of the sediments was generally warm temperate (warmer than the present-day climate of Poland), mild (without severe winters), and rather humid. Some fungal taxa (notably *Potamomyces* and probably *Tetraploa*) indicated a humid and warm, subtropical climate during this period [104,106,109].

Miocene sediments in Poland have a relatively rich palynological documentation, but most studies come from the Polish Lowlands [110,111]. In contrast, palynological (spore–pollen) studies of the Carpathian Foredeep are rare. Moreover, Badenian–Sarmatian marine sediments that were previously analysed were mainly from the western, Silesian region of the Carpathian Foredeep [110,112–115]. In addition, palynofloras from several boreholes in the Bochnia and Wieliczka regions [116], including borehole Kłaj 1 [117], as well as from the sulphur deposits at Piaseczno near Tarnobrzeg [102], were studied. Although the frequency of sporomorphs in these deposits was usually low, the samples usually had similar compositions, exhibiting abundant conifers with high levels of *Pinus* as well as *Abies*, *Tsuga*, and *Picea*. Among the deciduous trees, *Quercus*, *Ulmus*, *Castanea*, *Engelhardia*, and *Fagus* played the most significant roles, whereas *Carya*, *Pterocarya*, and *Tricolporopollenites pseudocingulum* (in some cases identified as *Rhus*) were less important. Shrubs and thermophilous ferns were also relatively frequent. Swampy plants, which are characteristic of continental sediments in the Polish Lowlands, were rare (except for *Taxodium*/*Glyptostrobus*) and included only taxa, such as *Alnus*, *Liquidambar*, *Myrica*, and *Ilex* [103].

The Babczyn palynoflora is very similar to the spore–pollen assemblage from the Jamnica S-119 borehole [103] drilled in the upper Badenian and lower Sarmatian marine deposits near Tarnobrzeg in the northeastern part of the Carpathian Foredeep. The Jamnica palynoflora was dominated by coniferous trees, especially *Pinus*, as well as *Tax-*

*odium/Glyptostrobus* (identified as Taxodiaceae–Cupressaceae), *Tsuga*, *Abies*, *Picea*, *Cedrus*, and *Sciadopitys*, whereas the pollen of *Sequoia* was identified only in some samples. Among deciduous trees and shrubs, *Ulmus*, *Quercus*, *Alnus*, *Carya*, *Fagus*, Ericaceae, *Engelhardia*, *Pterocarya*, and *Quercoidites henricii* were the most frequent. Some samples frequently contained small percentages of tree and shrub taxa, such as *Betula*, *Carpinus*, *Liquidambar*, *Salix*, *Acer*, *Castanea*, *Fraxinus*, *Juglans*, *Nyssa*, *Parrotia*, *Myrica*, *Symplocos*, Cyrillaceae/Clethraceae, *Ilex*, and *Tricolporopollenites pseudocingulum*. Pollen grains of herbaceous plants were very rare, with sporadic Chenopodiaceae, Cyperaceae, Poaceae, Lamiaceae, and *Nuphar*. Similarly, the spores of ferns (Polypodiaceae, *Osmunda*, and Cyatheaceae–Schizaeaceae) and *Sphagnum* appeared in very small quantities. The composition of plant communities in the entire Jamnica section was quite homogeneous, and no temporal flora changes were observed. The highest content of pollen material was in the middle part of the Jamnica profile, with more abundant pollen from deciduous trees and herbaceous plants, accompanied by a simultaneous decrease in *Pinus*, which indicated the more landward position of that part of the profile. Differences in the pollen spectra from Jamnica may suggest a shoreline migration at this level rather than changes in palaeovegetation caused by the climate [103].

Ref. [118] made similar assumptions for the entire Carpathian Region. Their palaeofloristic and palaeoclimatic reconstructions showed that the upper Badenian and lower Sarmatian floras were closer to each other than to the flora of the previous and subsequent stages. The evolutionary process during the late Badenian–early Sarmatian continued the main trends of forest floral and vegetational evolutions, stimulated by the gradual cooling of the climate [119]. At this time, the replacement of the subtropical forest communities by warm temperate and temperate forest communities was observed in the Central Paratethys. In the southeastern part of Ukraine (Eastern Paratethys), temperate forests were replaced by herb communities. This process was evident in all the studied sections, albeit involving spatial differences [118,120,121].

### 5.3. Implications

Although a consensus exists that at the Badenian–Sarmatian boundary, a sudden decrease in biodiversity and the disappearance of marine foraminifer assemblages in the Central Paratethys occurred (e.g., [122]), there are various hypotheses regarding its drivers. Ref. [123] concluded that this change occurred within a time interval of fewer than 10 kyr and that it was related to a change in the configuration of the Central–Eastern Paratethys gateway (Bârlad Strait). It could also have been related to a minor rise in the global sea level that otherwise affected the Sarmatian transgression in many other basins of the Central Paratethys ([124], with references therein). Ref. [125] suggested a continuous sea level rise in the Eastern Paratethys because of freshwater input by rivers, and, thus, the Sarmatian transgression event could have resulted from a full connection between both Paratethyan basins. It was repeatedly suggested that the transgression was accompanied by a drastic change in water chemistry, exemplified by a supposed change from marine to brackish conditions in the Central Paratethys (cf. [16,126]), which otherwise could be expected considering the inflow of Eastern Paratethyan water to the Central Paratethys.

Ref. [16] concluded that tectonic widening and the deepening of the Bârlad Strait generated an effective water exchange between the two domains. However, our data do not support the brackish-water environments at the onset of the Sarmatian, as envisaged by [16,123,127]. Instead, the domination of the foraminiferal assemblages above the Badenian/Sarmatian boundary by miliolids, which is commonly observed in various parts of the Carpathian Foredeep Basin in Poland [8,14], implies from normal to hypersalinity conditions lower than 50 psu [29,128]. As far as the palynological record on the onset of the Sarmatian is concerned, there are no blooms, or increased frequencies, of dinoflagellate cysts that would benefit from such environmental conditions, like euryhaline *Lingulodinium machaerophorum* or protoperidinioids. The former species is common in low-salinity waters (e.g., [129]) frequently found in recent sediments of the Black and Caspian Seas [130]. There are no freshwater algae, like *Botryococcus* and *Pediastrum*. On the contrary, samples from

the *Elphidium angulatum* Zone yielded blooms of Prasinophyta *Leiosphaera*, which is known to occur in the Middle Badenian strata associated with hypersaline conditions. Moreover, sample 33 yielded the highest proportion of *Polysphaeridium* (Figure 9), a genus commonly associated with increased salinity [95].

Dinocyst degradation can be excluded, as the same samples yielded other aquatic palynomorphs. The co-occurrence of *Elphidium* in those samples can be explained by a larger (compared to samples for the palynological study) rock volume taken from the core, which could also include relatively thin, possibly bioturbated layers with benthic communities that appeared during short periods of better sea bottom ventilation.

The authors in [131] concluded that from fully marine to hypersaline conditions existed in the late Sarmatian, and our results suggest that this conclusion is also valid for the early Sarmatian. This conclusion is based on the foraminiferal and aquatic palynomorph records (Figure 11). Warm and humid climatic conditions indicated by pollen obviously are not in favour of hypersalinity. But this may be related to warm/cold current circulations (there is evidence of such changes in intervals where cold-water *N. labyrinthus* interlayers with warm-water *Batiacasphaera*). Moreover, hypersaline conditions could appear independent of climatic conditions, for example, in the cases of basin isolation, the temporary cessation of circulation, and the appearance of stratification. Spore–pollen assemblages record changes in vegetation on land and changes in marine salinity, which will not be evident in these assemblages unless they were caused by a very pronounced change in the climate. Ref. [132] recorded only a slight cooling, and a reduction in humidity was marked at 14–13.5 Ma. No clear change was observed there, only fluctuations with a slight trend, despite an important environmental change related to the mid-Badenian Salinity Crisis [23,133]. The pollen record from marine sediments is not as precise as that from terrestrial/freshwater sediments, e.g., from lignite seams, which is something to be aware of. In the case of Babczyn, the spore–pollen assemblages are not very rich but are sufficient to draw some conclusions. Possible climatic fluctuations here could have been masked by changes in pollen transport and preservation conditions (i.e., water chemistry). Significant climate changes are not evident here though.

## 6. Conclusions

The qualitative and quantitative analyses of the foraminifera and the aquatic paly-nomorphs in the Badenian–Sarmatian strata of the Babczyn 2 borehole, one of the key sections in SE Poland, showed that the studied interval accumulated under variable, unstable sedimentary conditions in marine environments.

The dominance of infaunal benthic foraminifera species, mainly buliminids and uvigerinids, throughout the upper Badenian part of the studied interval (samples from 8 to 28) indicates mostly dysoxic and suboxic conditions. For the uppermost Badenian, mesotrophic conditions in surface waters and increased oxygenation of the bottom waters were observed. The Badenian/Sarmatian boundary, as correlated with a sudden extinction of stenohaline foraminifera, is placed just above sample 28 (within the lowermost part of the Syndesmya Beds). The overlying sample (29) is almost barren of foraminifera, and the lack of benthic forms may be explained by dysoxic conditions at the bottom of the sea. Euryhaline early Sarmatian benthic foraminiferal assemblages indicate a salinity increase, while the lack of planktonic foraminifera suggests a shallowing of the sea.

An acme of *N. labyrinthus* just below the Badenian/Sarmatian boundary may be the result of a temperature drop in sea-surface waters. The interval that occurred just above (samples 29 and 31) was characterised by a lack of dinoflagellate cysts and the flowering of Leiosphaeridiaceae, suggesting hypersaline conditions associated with stagnant, possibly stratified waters that led to anoxic conditions in the bottom waters, as evidenced by AOM. The subsequent cessation of these hypersaline conditions was caused by a possible sea-level rise and the gradual return of a less-saline water regime. The higher interval (samples 35–55) was deposited under more restricted and variable environmental condi-

tions, including a brackish environment, as indicated by diverse fungal microremains found in the upper part of the profile.

The results of the spore–pollen analysis revealed the presence of mesophytic and wetland vegetation along shoreline of the Paratethys during sedimentation and indicated that the climate during the deposition of the sediments was generally warm temperate, mild, and rather humid.

**Author Contributions:** Conceptualisation, D.P. and T.M.P.; methodology, D.P., P.G., E.W. and G.W.; validation, D.P., P.G., E.W. and G.W.; formal analysis, D.P., P.G., E.W., G.W. and T.M.P.; investigation, D.P., P.G., E.W., G.W. and T.M.P.; resources, D.P.; data curation, D.P., P.G., E.W. and G.W.; writing—original draft preparation, D.P., P.G., E.W., G.W. and T.M.P.; writing—review and editing, T.M.P., P.G., D.P., E.W. and G.W.; visualisation, D.P., P.G., E.W., G.W. and T.M.P.; supervision, D.P.; project administration, D.P.; funding acquisition, D.P. All authors have read and agreed to the published version of the manuscript.

**Funding:** This research was funded by the National Science Centre, Poland, grant number UMO-2017/27/B/ST10/01129.

**Data Availability Statement:** The data presented in this study are available on request.

**Conflicts of Interest:** The authors declare no conflicts of interest. The funders had no role in the design of the study; in the collection, analyses, or interpretation of the data; in the writing of the manuscript; or in the decision to publish the results.

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
