# Peer review of "Foraminiferal and Palynological Records of an Abrupt Environmental Change at the Badenian/Sarmatian Boundary (Middle Miocene): A Case Study in Northeastern Central Paratethys"

_geosciences, doi:10.3390/geosciences14030086_

Round 1

Reviewer 1 Report

Comments and Suggestions for Authors

I revised the manuscript of Peryt et al. “Foraminiferal and palynological record of an abrupt environmental change at the Badenian/Sarmatian boundary (Middle Miocene): a case study in northeastern Central Paratethys”, submitted to Geosciences-MDPI.

In this interesting paper, the authors present a foraminiferal and palynological investigation of some samples from a borehole section located in the north-eastern part of the Carpathian Foredeep (Poland). The aim of this study was to identify environmental changes around the Badenian/Sarmatian boundary (Middle Miocene). For this purpose, micropaleontological (51 samples) and palynological (26 samples) analyses have been carried out. In each sample, up to 200 benthic and planktonic foraminifera were collected, identified, and used for a paleoenvironmental reconstruction. Palynological analyses have been focused on dinoflagellate cysts, palynofacies (e.g., terrestrial pollen), and sporomorphs. All this has allowed to reconstruct the environmental changes that have affected the Central Paratethys around late Badenian and early Sarmatian, with the transition from an open marine environment to brackish conditions.

Overall, the manuscript is original, well organised and written, and provide an advancement of the current about the Paratethys faunal turnover event of Middle Miocene.

The methodology is generally correctly explained and applied. However, it is not clearly written how the nine benthic foraminiferal assemblages described in the paragraph 3.1 (page 10 and Figure 7) were identified. Were they likely identified based on the most abundant species present in each sample? In any case it must be written in the Methodology.

Results are described in detail and are reproducible, the data are interpreted appropriately; discussion and conclusions are supported by data and consistent with them.

The illustrations are of good quality (very nice the tables of foraminifera, dinoflagellate cysts and terrestrial palynomorphs) as well as the Tables are appropriate and easy to interpret and understand. The bibliography is wide and not include an excessive number of self-citations. The data availability statements are adequate.

Specific remarks are as follow:

-  page 2 line 52: Velapertina (= Praeorbulina) is incorrect. See WORMS site (https://www.marinespecies.org) and: Kiss et al. (2023). Convergent evolution of spherical shells in Miocene planktonic foraminifera documents the parallel emergence of a complex character in response to environmental forcing. Paleobiology, 49: 454-470.

- page 3 lines 112-118: some taxa are indicated as indicative of oxic, suboxic or dysoxic environments. Provide some specific references.

Moreover, I have doubts about the assignment of some taxa to groups oxic, suboxic and dysoxic. For example: Valvulineria complanata (included in the oxic group) is suboxic; Globocassidulina subglobosa (included in dysoxic group) is oxic; Uvigerina spp. (includes in dysoxic group) are suboxic. You can consult, among others:

Kaiho (1999). Effect of organic carbon flux and dissolved oxygen on the benthic foraminiferal oxygen index (BFOI). Marine Micropaleontology, 37: 67-76.

Kranner et al. (2022). Calculating dissolved marine oxygen values based on an enhanced Benthic Foraminifera Oxygen Index. Scientific Reports, 12, 1376.

Cavaliere et al. (2023). Paleoenvironmental Changes in the Gulf of Gaeta (Central Tyrrhenian Sea, Italy): A Perspective from Benthic Foraminifera after Dam Construction. Water, 15, 815.

- page 4, figure 1: in this figure, I would add the current position of the study area as in Figure 1 of Peryt et al. (2021), Geological Quarterly.

Other observations and details are listed in the attached pdf.

In conclusion, I think that the paper is suitable for the journal, after minor revisions.

Best Regards

Author Response

Thank you very much for your helpful comments. All your remarks (see below and also those in the annotated file) are accepted and the manuscript was accordingly modified except of one suggestion [figure 1: in this figure, I would add the current position of the study area as in Figure 1 of Peryt et al. (2021), Geological Quarterly], however, we have changed the graphical presentation of the location of the Babczyn 2 borehole.

Regarding your specific remark: The methodology is generally correctly explained and applied. However, it is not clearly written how the nine benthic foraminiferal assemblages described in the paragraph 3.1 (page 10 and Figure 7) were identified. Were they likely identified based on the most abundant species present in each sample? In any case it must be written in the Methodology., the appropriate information is now put in page 4, lines 134-135.

Your specific remarkS;

- page 2 line 52: Velapertina (= Praeorbulina) is incorrect. - We apologize for the mistake (no Praeorbulina now)

- page 3 lines 112-118: some taxa are indicated as indicative of oxic, suboxic or dysoxic environments. Provide some specific references. references added - page 4, lines 121-128

Moreover, I have doubts about the assignment of some taxa to groups oxic, suboxic and dysoxic. For example: Valvulineria complanata (included in the oxic group) is suboxic; Globocassidulina subglobosa (included in dysoxic group) is oxic; Uvigerina spp. (includes in dysoxic group) are suboxic - text in page 4 and 25 (changes are in yellow) is changed appropriately - thank you very much.

Reviewer 2 Report

Comments and Suggestions for Authors

The manuscritp is presented in a proper form.

NTRODUCTIONWell-structured and written.METHODSWell-structured and written. The description of the methods is clear.DISCUSSIONWell structured and well supported by the results. 

REFERENCES

To improveGENERAL COMMENTSOverall the work is very interesting and easy to read. Just some figures are of low quality. I have attached some more detailed comments in the attached pdf. file

Comments on the Quality of English Language

I suggest a bit more revision of the English. I am not mother tongue, but I found it in many parts “not fluid”. 

Author Response

Thank you for your helpful comments. In the revised version your remarks and comments referring to lines 119, 123, 125, 148,190,257,302, 314, 331, 554, 556, 577 and 605 were accepted and the text was appropriately corrected (all the changes are shown in yellow). Similarly, in Fig. 1 sample numbers were added on stratigraphic column, in Fig. 3 letters I,K,L were jointed, and a number of editorial erroers was eliminated (including double space in line 416).

Reviewer 3 Report

Comments and Suggestions for Authors

Dear authors,

the object of your research is very interesting as a debate is still open in the scientific community. Although, some alternative datatin methods would be welcome, the multi-fossil approach is very useful. We used this method for a Messinian succession in the Mediterranean (Bertini et al., 2024) and comparing calcareous with organic walled microfossil is important because they have different taxonomic history and pollen can allow a comparison of the marine and continental record.

However, I found the manuscript a bit difficult to read. This observation originates from the organization of the paper and the information in each chapter.

The Introduction is rather long and has too many detail, which would fit better in the geological background or in the discussion.

The methods are described in very different ways. Forams are very short, while palynomorphs are described extensively. A balance is required.

In the results I found difficult to follow the many assemblages you described regarding the forams and the detailed, sample scale of the palinomorph. I think you should try to find out larger group of samples to describe although they can be a little heterogeneous and focus your attention on trends.

The discussion is very long and is not focussed on the environmental evolution, but rather on the different group separately. Focussing on the evolution (you can subdivide the long chapter into shorter ones, each characterized by an homogeneous setting). A figure would undoubtedly help the reader to follow the discussion.

The implication chapter is rather short. I do not understand if you agree with Palcu or with Filipescu and Silye or you have a third opinion. Do you think the boundary is coeval to that studied by other authors in other part of the Paratethys (Palcu, Filipescu, etc.)? Or is it time transgressive?

So, I think major revisions are required before publication of your valuable dataset. I commented in the text what expressed above and other minor things.

all the best

Comments on the Quality of English Language

I am not English native language, but I think the text require a moderate revision of the syntax. Some words and some sentences are not correct and need to be revised.

Author Response

Dear Reviewer,

Thank you very much for your comments and remarks. We have been trying to modify our paper following your suggestions (and that is why we have ask for one week more for the revision) but finally we ended up with a slightly modified original plan of our paper. 

You indicate that The Introduction is rather long and has too many detail, which would fit better in the geological background or in the discussion. We added a new chapter (Geological setting).

You correcty state that The methods are described in very different ways. Forams are very short, while palynomorphs are described extensively. A balance is required. However, the balance would indicate, in our case, that the foram part would be enlarged, and we see no reason for that. This chapter was slighly modified regarding the foram part only, therefore.

In the results I found difficult to follow the many assemblages you described regarding the forams and the detailed, sample scale of the palinomorph. I think you should try to find out larger group of samples to describe although they can be a little heterogeneous and focus your attention on trends.

The discussion is very long and is not focussed on the environmental evolution, but rather on the different group separately. Focussing on the evolution (you can subdivide the long chapter into shorter ones DONE, each characterized by an homogeneous setting). A figure would undoubtedly help the reader to follow the discussion. New figure is added (Fig. 11) - thank you for your very helpful suggestion.

The implication chapter is rather short. It is slightly enlarged. I do not understand if you agree with Palcu or with Filipescu and Silye or you have a third opinion. Do you think the boundary is coeval to that studied by other authors in other part of the Paratethys (Palcu, Filipescu, etc.)? Or is it time transgressive? We think that Badenian/Sarmatian boundary (=BSEE) was coeval.

So, I think major revisions are required before publication of your valuable dataset. I commented in the text what expressed above and other minor things.

Round 2

Reviewer 3 Report

Comments and Suggestions for Authors

Dear authors,

I appreciate your efforts. I still see many problems in your manuscript. Results are presented in high detail. It is not easy to understand and follow the general trend. You followed the suggestion to make assemblages, but these are sometimes formed by one single sample. In the discussion, these assemblages do not really represent intervals of the evolution that you reconstruct. The reader has to link the interpretation of the different that groups by himself and check if they are consistent. For example the hypersalinity reconstructed by means of miliolids is not supported by climatic conditions (warm and humid); same for the cooling, there is no sign of cooling in the pollen. Also is difficult to reconcile stratification with varying salinity and depth.

In general, I suggest to shorten the discussion about the single group, which is very difficult to follow and somehow bear poor significance and strengthen the implications, which should take into account all the data. The different data should as much as possible tell the same story and all your inferences should be supported by data.

I wrote comments in the text to support these criticisms in order to help you revise the manuscript.

best regards

Author Response

Dear Reviewer,

Our concept of the paper is to present a detailed dataset based on a study of different groups (each has various resolution and limitations), and hence the conclusions may differ, instead of telling one story through selecting the data. We enlarged now the implications chapter (new text is shown in yellow) including our answer to your general comment (For example the hypersalinity reconstructed by means of miliolids is not supported by climatic conditions (warm and humid); same for the cooling, there is no sign of cooling in the pollen.) as well as your remark (can we exclude degradation of cysts? why elphidium are present if waters are stratified and bottom waters are anoxic?) to the paragraph starting at the previous line 696 (“The cessation of these…”).

Here are our answers to your specific comments (the lines refer to the previous version; new texts are highlighted in yellow).

Fig. 2M. You are right – we apologize for a mistake. However, the former Figure 3A was Trilobatus, although in the current Figure 3A we illustrate more convincing specimen.

Fig. 3B. The illustrated specimen is Globigerina and not Ciperoella: The latter taxon possesses a distinctive reticulate wall while Globigerina is characterized by spinose wall where spine collars often coalesce to form ridges.

Line 285 CORRECTED

Line 295 CORRECTED

Line 348 CORRECTED

Line 560 Trilobatus is shown in Fig. 3A

LINE 586 NOW USED IN DISCUSSION

Line 591 CORRECTED

LINE 601 EXPLAINED

LINE 621 CORRECTED

Line 627 CORRECTED

Line 640 CORRECTED

Line 653 [Moreover, Nematosphaeropsis labyrinthus occurred during this interval and exhibited a negative correlation with Batiacasphaera (Figure 9)]. what does it mean?

Answer: It means that frequencies of Nematosphaeropsis labyrinthus in this interval increase in samples with rare Batiacasphaera and they drop in samples that yielded frequent Batiacasphaera. As it is explained below in the text, such a correlation may be a result of water depth fluctuations, as N. labyrinthus is rather an offshore species, whereas Batiacasphaera is more proximal taxon. It may also reflect changes of surface water temperature: periods of warmer waters indicated by a warm-water Batiacasphaera and colder waters with N. labyrinthus.

Line 693 [These possible hypersaline conditions were associated with stagnant, possibly stratified waters that led to anoxic conditions in the bottom waters, as evidenced by AOM (Figure 8).] how can hypersaline water be stagnant? How much hypersaline?

Answer: Hypersaline waters are usually stagnant. If waters are affected by strong circulation, particularly in a deeper basin, then hypersaline conditions are less likely to appear. Palynological record shows blooms of Leiosphaera, a prasinophyte alga widespread in the Miocene evaporitess of the Carpathian Foredeep, simultaneously with appearance of amorphous organic matter (AOM) that is typical for anoxic bottom conditions. We have no indications as to precise salinity level (certainly above 3.5%).

Line 773 CORRECTED

Kind regards,

Tadeusz Peryt

Round 3

Reviewer 3 Report

Comments and Suggestions for Authors

Dear Authors,

I read your explanation and I realize that the change made to the manuscript have sufficiently increased the quality of the manuscript, so that it can be accepted for publication in the present form.

best regards